# Estimating the Probabilities of Rare Outputs in Language Models

**Gabriel Wu**[*]    **Jacob Hilton**
Alignment Research Center

## Abstract

We consider the problem of *low probability estimation*: given a machine learning model and a formally-specified input distribution, how can we estimate the probability of a binary property of the model's output, even when that probability is too small to estimate by random sampling? This problem is motivated by the need to improve worst-case performance, which distribution shift can make much more likely. We study low probability estimation in the context of argmax sampling from small transformer language models. We compare two types of methods: importance sampling, which involves searching for inputs giving rise to the rare output, and activation extrapolation, which involves extrapolating a probability distribution fit to the model's logits. We find that importance sampling outperforms activation extrapolation, but both outperform naive sampling. Finally, we explain how minimizing the probability estimate of an undesirable behavior generalizes adversarial training, and argue that new methods for low probability estimation are needed to provide stronger guarantees about worst-case performance.

## 1 Introduction

Modern ML systems undergo black-box optimization to minimize a loss function on samples drawn from a training distribution. Although models produced in this way perform desirably on average over this distribution, they can still produce highly undesirable outputs on very rare inputs. This is a problem, because these rare inputs can become much more likely in the presence of distribution shift, especially one chosen adversarially, such as with large language model "jailbreaks" (Carlini et al., 2024; Wei et al., 2024).

Preventing such highly undesirable outputs is a notoriously challenging problem. The most common remedy is adversarial training, in which inputs that produce these undesirable outputs are searched for and used as additional training data (Goodfellow et al., 2014; Madry, 2017), but the transfer between different search methods is generally weak (Kang et al., 2019; Wei et al., 2024). In this work, we propose the more modest goal of simply *estimating the probability* that an input drawn from some distribution will produce a certain kind of output, which has been considered before in the context of computer vision in Webb et al. (2019). We will show that even this intermediate goal is challenging, but successful methods could enable new ways of preventing undesirable outputs by minimizing their estimated probability.

To advance work on this problem, we study low probability estimation in the context of small transformer language models. We consider various formally-defined input distributions in which each input token is sampled independently, and develop methods for estimating the probability that a particular target token will have the largest output logit. We constrain the computational budget of our methods and obtain ground truth probabilities by random sampling using a much larger computational budget. The target tokens are chosen to have ground truth probabilities between $10^{-9}$ and $10^{-5}$, which are too small for random sampling to produce a good estimate under the constrained computational budget.

In this context, we study two types of methods:

---

[*]Correspondence to: `gabriel.d.wu314@gmail.com`

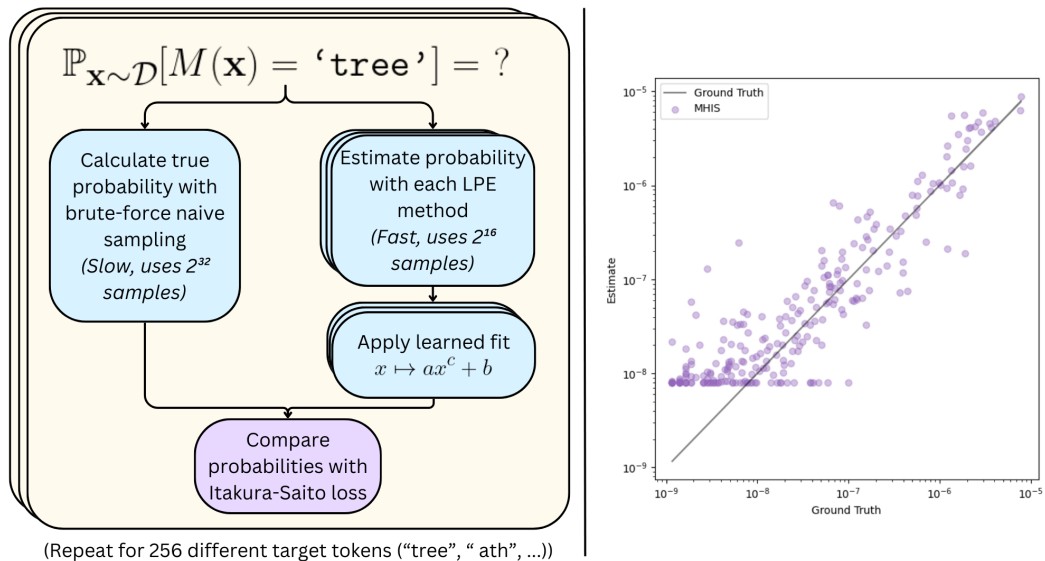

Figure 1: *Left*: To evaluate the performance of our low probability estimation methods, we compare their estimates against ground-truth probabilities obtained by brute-force sampling with a larger computational budget. *Right*: The estimates of Metropolis–Hastings Importance Sampling on the `icl` input distribution and 4-layer model, after a fit has been applied. Each point represents a different target token.

- **Importance sampling.** We define a new input distribution under which the rare event is much more likely, sample from that distribution, and re-weight samples to obtain an unbiased estimate for the original distribution. Our Independent Token Gradient Importance Sampling (ITGIS) method treats token positions independently and uses gradients to obtain this new input distribution, while our Metropolis–Hastings Importance Sampling (MHIS) method uses a Markov chain Monte Carlo algorithm to sample from a distribution with non-independent tokens.

- **Activation extrapolation.** We use random samples to fit a probability distribution to the model's logits, and extrapolate into the tails of this distribution to produce a probability estimate. Our Quadratic Logit Decomposition (QLD) method applies a presumption of independence to the empirical distribution of logits, motivated by Christiano et al. (2022), and our Gaussian Logit Difference (GLD) method is a simple baseline that fits a Gaussian to the difference between the maximum logit and target logit.

In our setting, both types of methods outperform random sampling, and importance sampling tends to outperform activation extrapolation. Nevertheless, we remain interested in activation extrapolation and similar approaches because they produce new methods for reducing the probabilities of rare outputs, whereas importance sampling essentially recovers standard adversarial training.

The remainder of the paper is structured as follows. In Section 2, we formally define the problem of low probability estimation, both in general and in our language model setting. In Section 3, we describe our four methods in more detail. In Sections 4 and 5, we describe the models and input distributions on which we test our methods and convey our experimental findings. Finally, in Sections 6, 7 and 8, we discuss the limitations and implications of our results, related work, and future directions.

## 2 PROBLEM STATEMENT

Given an input space $\mathcal{X}$, an output space $\mathcal{Y}$, an input distribution $\mathcal{D} \in \Delta(\mathcal{X})$, a model $M : \mathcal{X} \to \mathcal{Y}$, and a formal boolean property of model outputs $C : \mathcal{Y} \to \{0, 1\}$, low probability estimation is the

problem of efficiently estimating

$$\Pr_{\mathbf{x}\sim\mathcal{D}}[C(M(\mathbf{x})) = 1].$$

We sometimes refer to the event $C(M(\mathbf{x})) = 1$ as the "target behavior", or just "the behavior." If the probability of the behavior large enough (say, larger than $1/n$), it is easy to estimate by drawing $n$ independent samples from $\mathcal{X}$ and using the sample mean of $C(M(\mathbf{x}))$. However, if the probability is significantly smaller than $1/n$, this sample mean is almost always $0$, making it uninformative at distinguishing between small probabilities like $10^{-10}$ and $10^{-20}$.

## 2.1 OUR SETTING

In this paper, we study low probability estimation in the setting of argmax sampling from language models with single-token behaviors. Let $M : \mathcal{V}^* \to \mathcal{V}$ be a transformer language model that predicts the next token given a string of previous tokens, where $\mathcal{V}$ is the token vocabulary. Note that we sample at temperature $0$, so $M$ is deterministic. Given a distribution $\mathcal{D}$ over $\mathcal{V}^*$ and a target token $t \in \mathcal{V}$, the low probability estimation problem for single-token behaviors is the task of estimating

$$\Pr_{\mathbf{x}\sim\mathcal{D}}[M(\mathbf{x}) = t].$$

Letting $M_i(\mathbf{x})$ be the logit the model assigns to token $i \in \mathcal{V}$, this can also be written as:

$$\Pr_{\mathbf{x}\sim\mathcal{D}}[M_t(\mathbf{x}) > M_i(\mathbf{x}) \quad \forall i \neq t].$$

In general, $\mathcal{D}$ can be any distribution that can be formally specified. However, in this paper we focus only on distributions $\mathcal{D}$ with independent tokens. That is, we specify an input length $k$ and token distributions $\mathcal{D}_1, \ldots, \mathcal{D}_k \in \Delta(\mathcal{V})$, then write $\mathcal{D}$ as the product $\mathcal{D}_1 \times \cdots \times \mathcal{D}_k$. Table 1 shows the 8 distributions that were tested, with tokens colored for clarity. To prevent overfitting, the methods were only run on the first four distributions during development, and they were finalized before testing on the last four distributions. The results were qualitatively the same on both halves of the split.

Table 1: Input distributions and examples. See Table 2 for more detailed descriptions.

| Name | Short description | Tokenized example |
|------|-------------------|-------------------|
| hex | Hexadecimal characters | `<|BOS|>aa5acbf6aad468813f94c2fbbff4dc65eadc1553` |
| camel | CamelCase Python tokens | `<|BOS|>LayoutCredServicesVirtualUseTimeInterface ColorBodyAlRowHeightRepFontAndMetaRequestGroupsOne LabelPasswordAndRaVideoFailedValueGuiTypeMicrosoft SlotDeId` |
| colon | Python tokens, ending with ':' | `<|BOS|>    et-= """]: (\n       : Thisc \r\n                 ('/\nFilereturn\n\n         <|EOS|>␣ '].2default.**1  self( def')",:` |
| if | Python tokens, starting with ' if' | `<|BOS|> if: else,-post\n        \n        2\n\n5 found fromout, self- node +=\n \n       =\n( this 'values (),.(do` |
| caps | "He/She screamed:", followed by caps and punctuation | `<|BOS|>He screamed: "ESOTTULEBOV.,WR!!IMITLEER.,ARY ...IIESSION` |
| english | English words | `<|BOS|>ating. is invent School not found from cm an in one to shooting everyone Cor George around responsive employees ground on stone various,` |
| spanish | Spanish tokens | `<|BOS|> lo no bu dees cr socialjosabiler m de enidad areljd final de v de lo much` |
| icl | In-context learning prompt | `<|BOS|>A for American R for Return C for crack T for troubles H for house E for equipment O for operating R for reason Y for your V for` |

## 3 ESTIMATION METHODS

We introduce four methods in this section: two *importance sampling* methods (Independent Token Gradient and Metropolis–Hastings), and two *activation extrapolation* methods (Quadratic Logit Decomposition and Gaussian Logit Difference). We also compare against the baseline of outputting an

optimal constant, which can be thought of as the performance of naive sampling because we only evaluate the methods on tokens with ground truth probabilities less than the reciprocal of the allotted sampling budget (see Section 4).

## 3.1 IMPORTANCE SAMPLING METHODS

Naive sampling fails to produce good estimates for low probability events because it takes too many samples from $\mathcal{D}$ to observe a positive example. To address this, we can instead draw samples from a different distribution that up-weights regions of input space most likely to produce the behavior of interest. If we re-weight our observations properly, this gives an unbiased estimator for the true probability. This is known as *importance sampling*, and it enjoys the same advantages that adversarial training has over standard training: by using a narrower input distribution, we can more efficiently discover positive examples of the target behavior.

Formally, let $p(\boldsymbol{x})$ be the probability mass function of $\mathcal{D}$, and let $q(\boldsymbol{x})$ be the PMF of any other distribution. Then

$$\Pr_{\mathbf{x}\sim p}[M(\mathbf{x})=t] = \mathbb{E}_{\mathbf{x}\sim p}[\mathbb{1}[M(\mathbf{x})=t]] = \mathbb{E}_{\mathbf{x}\sim q}\left[\frac{p(\mathbf{x})}{q(\mathbf{x})}\mathbb{1}[M(\mathbf{x})=t]\right],$$

but the latter may have less variance (and so require fewer samples to get a good estimate).

The following two importance sampling methods take $q(\boldsymbol{x})$ to be a Boltzmann posterior with prior $p(\boldsymbol{x})$. The first defines $q(\boldsymbol{x})$ with independent tokens, while the second defines $q(\boldsymbol{x})$ to have non-independent tokens and so requires a more sophisticated sampling method.

### 3.1.1 INDEPENDENT TOKEN GRADIENT IMPORTANCE SAMPLING (ITGIS)

We want $q$ to up-weight tokens that contribute to $t$ being outputted. One way to do this is to continue to treat each input token as independent, but change the probability of tokens according to their average linear contribution to the logit of $t$. Let $\boldsymbol{x} = (x_1, \ldots, x_k) \in \mathcal{V}^k$ be an input of length $k$, and say that $p(\boldsymbol{x})$ factors as $p_1(x_1)\cdots p_k(x_k)$. Then we define $q(\boldsymbol{x}) = q_1(x_1)\cdots q_k(x_k)$, where

$$q_i(x_i) \propto p_i(x_i) \cdot \exp\left(\frac{s_i(x_i)}{T}\right)$$

and

$$s_i(x_i) = \mathbb{E}_{\mathbf{x}'\sim q}[\nabla_{\mathbf{x}'}M_t(\mathbf{x}')]_{i,x_i}.$$

$T$ is a temperature parameter, and the gradient is taken by treating $\mathbf{x}'$ as a one-hot vector in $\mathbb{R}^{k\times|\mathcal{V}|}$. Intuitively, the gradient $\nabla_{\mathbf{x}'}M_t(\mathbf{x}')_{i,x_i}$ gives us a linear approximation to how much the logit of $t$ would change if we replaced $i$-th token of $\mathbf{x}'$ with $x_i$ (up to an additive constant w.r.t. $x_i$). Thus, $s_i$ scores each token value according to its average linear contribution to $M_t$, and $q_i$ is defined as the Boltzmann distribution with respect to this score function.[1]

However, since $s_i$ and $q$ are both defined in terms of each other, we can't calculate $s_i$ directly. To overcome this, we construct a sequence of score functions $s_i^{(0)}, s_i^{(1)}, \ldots$ and associated distributions $q^{(0)}, q^{(1)}, \ldots$ that are adaptively refined with respect to each other (see Appendix B.i for details). Sampling from each $q^{(j)}$ lets us calculate an importance sampling estimate, and the final output of the method is the average value of these estimates across all $j$.

### 3.1.2 METROPOLIS–HASTINGS IMPORTANCE SAMPLING (MHIS)

A problem with ITGIS is that the new sampling distribution $q(\boldsymbol{x})$ still treats all tokens as independent, and it only accounts for linear effects of tokens on the target logit. Thus, ITGIS may fail to sample into the most important regions of the input space if the model is sensitive to non-linear interactions between tokens (e.g., if the model's target logit is only high when the last two tokens of the input are the same as each other).

---

[1]It can be shown that, given a score function $s(x)$ and a prior $p(x)$, the distribution that maximizes $\mathbb{E}_{\mathbf{x}\sim q}[s(\mathbf{x})] - T \cdot \mathrm{KL}(q\|p)$ is $q(x) \propto p(x) \cdot \exp(s(x)/T)$.

To remedy this, we can define an importance sampling distribution that doesn't have independent tokens. We must use a score function that depends on the entire input; the most natural choice is the target logit $M_t(\boldsymbol{x})$. We define

$$q(\boldsymbol{x}) \propto p(\boldsymbol{x}) \cdot \exp\left(\frac{M_t(\boldsymbol{x})}{T}\right),$$

again using a Boltzmann distribution to up-weight regions of input space that are more likely to have positive samples.

Unlike ITGIS, we cannot explicitly compute $q$ because it does not factor into independent distributions over each token. Instead, we use the Metropolis–Hastings algorithm to produce a random walk in input space that has a stationary distribution of $q$.[2] To do so, we must define a proposal distribution $\phi(\boldsymbol{x}'|\boldsymbol{x})$ that suggests the next element of the walk. To encourage fast mixing, this proposal distribution should be good at exploring into regions of input space that $q$ weights highly.

Here we take inspiration from Greedy Coordinate Gradient, an algorithm that optimizes a discrete prompt to jailbreak a model using gradients (Zou et al., 2023). We adapt this optimization procedure into a proposal distribution: to pick a proposed next step $\mathbf{x}'$ of the walk, we choose a random token position $i$ to replace, compute the gradient of $s(\mathbf{x})$ with respect to $\mathbf{x}_i$, then sample a replacement token for position $i$ according to a Boltzmann distribution defined by this gradient (similarly to ITGIS). The final output of the method is the average importance sampling estimate taken after a burn-in period. For a precise description of the algorithm, see Appendix B.ii.

### 3.2 ACTIVATION EXTRAPOLATION METHODS

The importance sampling methods search for explicit examples of inputs that cause the given behavior. This makes their task at least as hard as the adversarial training search problem—if it is difficult to find an $\boldsymbol{x} \in \mathrm{supp}(\mathcal{D})$ such that $M(\boldsymbol{x}) = t$, the importance sampling estimators will likely fail to produce a positive estimate.

We hope to find low probability estimation methods that work even when the search problem for importance sampling is hard. To do this, we introduce activation extrapolation: first fit a distribution to the activations or logits of $M$, then estimate the probability of the output property of interest under this idealized distribution. Our first such method is Quadratic Logit Decomposition, which applies a presumption of independence between uncorrelated subspaces the model's pre-unembed activations. We also develop Gaussian Logit Difference, which is intended as a simple baseline method.

#### 3.2.1 QUADRATIC LOGIT DECOMPOSITION (QLD)

Let the random vector $\mathbf{v}(\mathbf{x}) \in \mathbb{R}^d$ be the activation of the model right before applying the unembed matrix $\boldsymbol{W}_U \in \mathbb{R}^{d \times |\mathcal{V}|}$. That is, $\mathbf{v}(\mathbf{x}) \cdot \boldsymbol{W}_U$ represents the model's output logit vector $M(\mathbf{x})_{1,\ldots,|\mathcal{V}|}$. We first collect $n$ samples of $\mathbf{v}$ (call them $\mathbf{v}^{(1)}, \ldots, \mathbf{v}^{(n)}$). We then choose some unit direction $\boldsymbol{d} \in \mathbb{R}^d$ (see below), then decompose each $\mathbf{v}^{(i)}$ into $\mathbf{a}^{(i)} + \mathbf{b}^{(i)}$, where $\mathbf{a}$ lies in the subspace spanned by $\boldsymbol{d}$, and $\mathbf{b}$ lies in the complementary subspace that is orthogonal in a whitened basis.[3] This decomposition is chosen such that the random vectors $\mathbf{a}$ and $\mathbf{b}$ are uncorrelated across the $n$ samples.

Next, by treating the random vectors $\mathbf{a}$ and $\mathbf{b}$ as independent, we can use our $n$ samples of each to obtain $n^2$ "synthetic" samples of $\mathbf{u}$. The final output of QLD is the proportion of these synthetic samples that cause $t$ to be outputted:

$$\frac{1}{n^2}\left|\left\{(i,j) \in [n]^2 \mid \mathbf{a}^{(i)} + \mathbf{b}^{(j)} \in S\right\}\right|,$$

where $S \subseteq \mathbb{R}^d$ is the "acceptance region" of activation space corresponding to activations that result in the target logit being highest after unembedding. Despite the fact that there are $n^2$ synthetic samples, this proportion can be computed in $\tilde{O}(n)$ time by first sorting the samples $\mathbf{a}^{(i)}$. A more complete description of the QLD algorithm can be found in Appendix B.iii.

---

[2]Metropolis–Hastings is a Markov Chain Monte Carlo method for sampling from a distribution with an unknown normalizing constant. See Robert (2016) for a description of the algorithm.

[3]Actually, $\mathbf{a}$, $\mathbf{b}$, and $\boldsymbol{d}$ are all defined in the whitened space of $\mathbf{v}$; see Appendix B.iii.

**Choice of direction.** We rely on the following two assumptions for QLD to perform well: 1) $\mathbf{a}$ and $\mathbf{b}$ are independent (so that our estimate is unbiased), and 2) the contribution towards the output behavior is split roughly equally between these two terms (to minimize the variance of our estimate). See Appendix C for more discussion of this motivation. After some initial experimentation with a variety of candidate directions,[4] we decided to set $\boldsymbol{d}$ to be the direction of the shortest vector in whitened space that results in the model outputting $t$. It can also be thought of as the maximum likelihood value of $\mathbf{v}$ under a Gaussian prior, conditioned on observing the model output $t$. Appendix D describes the algorithm we use to compute $\boldsymbol{d}$.

### 3.2.2 GAUSSIAN LOGIT DIFFERENCE

On any given input, we can record the difference $\Delta_t := M_t(\mathbf{x}) - \max_i M_i(\mathbf{x})$. We wish to estimate the probability that $\Delta_t \geq 0$. A natural estimation method, which we view as a simple baseline, is to treat $\Delta_t$ as Gaussian by estimating its mean $\mu$ and standard deviation $\sigma$ with samples, then calculate $\Pr[\mathcal{N}(\mu, \sigma^2) \geq 0]$. In practice, we use a slightly different functional form that captures the Gaussian PDF, which approximates the CDF well in the tails. The output of the Gaussian Logit Difference method is:

$$\exp\left(-\left(\frac{a\mu}{\sigma + \epsilon}\right)^2 + b\right) + c,$$

where $a, b, c$, and $\epsilon$ are parameters that are fit to minimize loss across all target tokens associated with a given distribution (see Section 4.2).

## 4 EXPERIMENTAL SETUP

We apply our methods on three models: a 1-layer, a 2-layer, and a 4-layer transformer from Nanda & Bloom (2022). All models have a hidden dimension of $d = 512$, a vocabulary size of $|\mathcal{V}| = 48262$, GELU non-linearities (Hendrycks & Gimpel, 2023), and were trained on the C4 dataset (Raffel et al., 2023) and CodeParrot (Tunstall et al., 2022).

For each of the 8 distributions (listed in Table 1) and for each model, we generate ground-truth token probabilities by running forward passes on $2^{32}$ random samples. We then select a random set of 256 tokens among those with ground-truth probabilities between $10^{-9}$ and $10^{-5}$, and we test all of our methods on these tokens.

We give each method a computational budget of $2^{16}$ model calls (see details in Appendix F). This budget was chosen so that naive sampling would almost never result in any positive estimates for the range of token probabilities we test ($2^{16} < 10^5$), but the theoretical quadratic gains from QLD would still be enough to get signal on the entire range of probabilities ($(2^{16})^2 > 10^9$).

Our code is available at `https://github.com/alignment-research-center/low-probability-estimation`.

### 4.1 ITAKURA–SAITO LOSS

We measure the quality of the method with a loss function inspired by the Itakura–Saito divergence (Itakura & Saito, 1968). If $p$ is the ground-truth probability of a particular target token, then an estimate of $q$ incurs a loss of:

$$D_{\text{IS}}(p, q) = \frac{p}{q} - \ln \frac{p}{q} - 1.$$

Two considerations went into the choice of this loss function. First, Itakura–Saito loss is a proper scoring rule (Buja et al., 2019). Second, since it only depends on the ratio $p/q$, Itakura–Saito loss is sensitive to small probabilities: if $p = 10^{-100}$ and $q = 10^{-10}$, then $D_{\text{IS}}(p, q)$ is very large. In contrast, the squared error loss function $(p - q)^2$ would be extremely small. Intuitively, this sensitivity is desirable because we care how our methods perform on a wide (as measured in log-space) range of ground-truth probabilities. We don't want the performance metric to be dominated by a method's behavior on only the most probable tokens.

---

[4]Other candidate directions included 1) the $t$-th column of $\boldsymbol{W}_U$ pulled back into whitened space and 2) the expectation of $\mathcal{N}(0, \text{Id}_d)$ conditioned on lying in the whitened acceptance region.

For completeness, we also report our results using squared error in log-space (Appendix H), even though this is not a proper scoring rule. The results are qualitatively identical.

## 4.2 AFFINE FITS

Many methods often report estimates of 0, but $D_{\text{IS}}$ is undefined for $q = 0$. To address this, we fit a transformation $x \mapsto ax^c + b$ to the outputs of each method, where $a, b$ and $c$ are chosen to minimize Itakura–Saito loss. $ax^c$ can be thought of an affine transformation in log-space, and adding $b$ prevents values from being too small while barely affecting larger outputs. To ensure that this transformation is not overfitting to the particular set of 256 tokens, we report the leave-one-out cross-validation (LOOCV) loss of each method. We train a separate fit for each (method, input distribution) pair. [5]

## 5 RESULTS

Figure 2 shows the performance of each method. The relative ordering is clear: both importance sampling methods outperform Quadratic Logit Decomposition, which in turn outperforms Gaussian Logit Difference. GLD is barely better than outputting an optimal constant (which can be interpreted as the performance of naive sampling). Figure 4 shows that there is a fair amount of variation in method performance across the 8 distributions: some behaviors like `hex` and `icl` favor MHIS, while others like `spanish` heavily favor ITGIS. A more detailed table of results is in Appendix G.

Among the two importance sampling methods, ITGIS does better on smaller models, while MHIS does better on larger models. We believe this is because larger models are less easily approximated as linear functions and are more likely to have complex behaviors arising from inter-token interactions.

Figure 3 displays example scatter plots of ITGIS, MHIS, and QLD estimates before a fit is applied. Each point represents the ground-truth and estimated probability of a different target token. More scatter plots can be found in Appendix J; note that the qualitative performances of the methods can vary significantly on different input distributions. We perform an ablation study on our choice of loss function in Appendix H, in which we score methods based on squared error in log-space instead of Itakura–Saito loss.

## 6 DISCUSSION

### 6.1 DISTRIBUTION SHIFT AS MOTIVATION

One might ask: if a particular model behavior is so rare that it never arises during training, why would we care about estimating its probability? There are a few reasons. First, some AI systems may be run on many more inputs during the course of deployment than during training. Thus, if a certain model behavior would be so catastrophic that it is unacceptable for it to occur even once in deployment, we cannot rely on training to drive down its probability low enough. Second, there may be distributional shift between training and deployment such that events that occur extremely rarely during training become more likely in deployment. This could occur because of an input chosen adversarially, but it could also occur because of goal misgeneralization (Shah et al., 2022).

A particularly challenging case is *deceptive alignment*, the possibility that an ML model would look for clues about whether it is in a training or a deployment environment, and only behave well in training (Hubinger et al., 2021). To detect whether a model is deceptively aligned, one could craft an input distribution that is "wide enough" to assign *some* probability mass, even if very small, to any possible deployment-time input, then apply low probability estimation methods to detect if the model would ever perform a catastrophic behavior on this distribution.[6] For more discussion of this idea, see Xu (2024).

---

[5]Note that the Gaussian Logit Difference method has a special functional form of its fit $((\mu, \sigma) \mapsto \exp\left(-\left(a\mu/(\sigma + \epsilon)\right)^2 + b\right) + c$ instead of $x \mapsto ax^c + b$) but is otherwise evaluated in the same way.

[6]To prevent false positives, this would require a very demanding definition of catastrophe that would be impossible for the model to trigger "by accident."

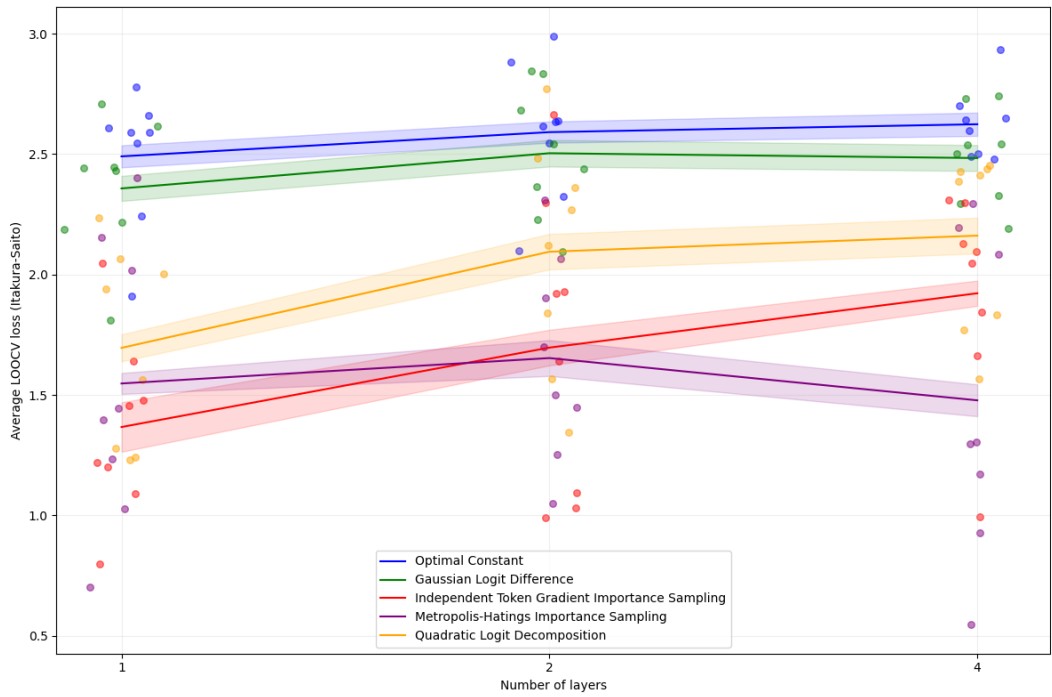

Figure 2: The Itakura–Saito loss of all methods across different model sizes. The solid lines indicate the loss of each method averaged over all 8 distributions, with bands showing standard error. The colored points indicate the loss on individual distributions, with horizontal jitter added for visibility. Lower is better.

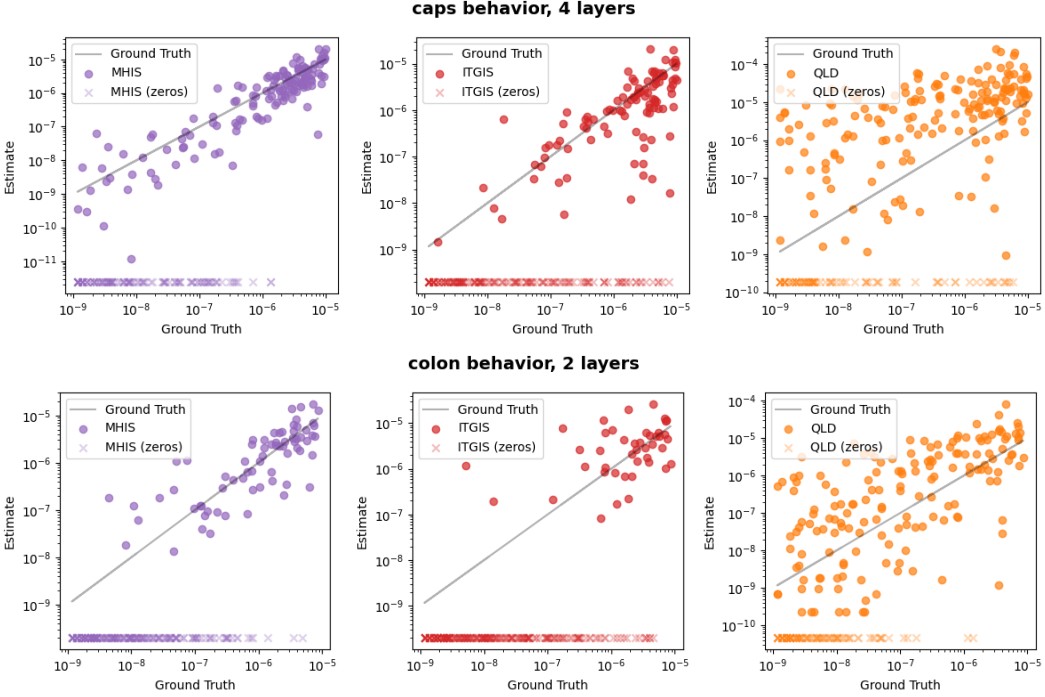

Figure 3: Examples of method outputs on two different behaviors and models, before a fit is applied. Estimates of 0 are placed at the bottom of each graph for visibility.

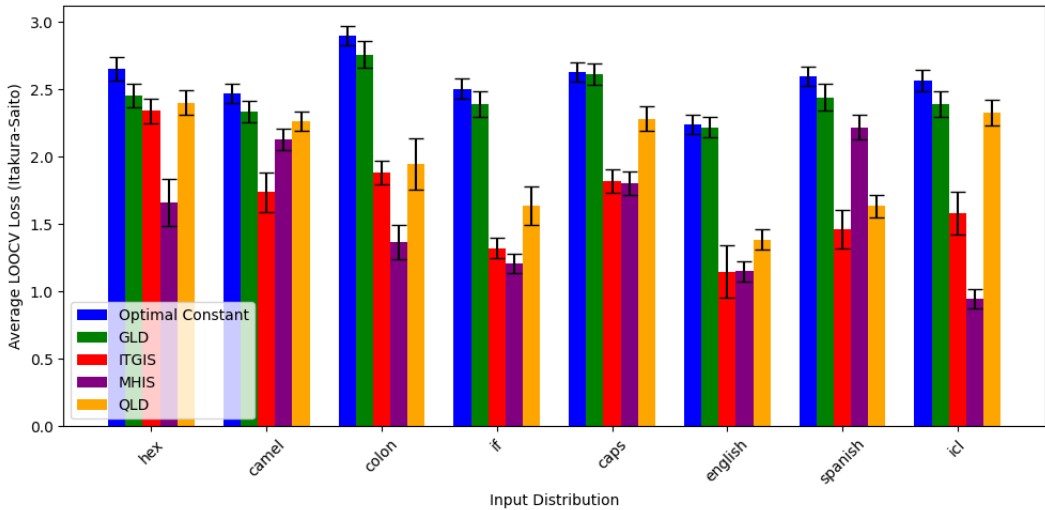

Figure 4: The Itakura–Saito loss of all methods across different distributions, averaged over all 3 model sizes. Lower is better.

## 6.2 RELATION TO RED-TEAMING AND ADVERSARIAL TRAINING

Our importance sampling methods for low probability estimation involve finding inputs for which the rare event occurs. This amounts to the well-studied task of "red-teaming". As long as the required importance sampling ratios can be computed, any method for red-teaming can be turned into an importance sampling method for low probability estimation, as we demonstrate with our adaptation of Greedy Coordinate Gradient into MHIS (Zou et al., 2023). However, our activation extrapolation methods such as QLD do not correspond to any red-teaming method.

A further reason to be interested in low probability estimation is that it could be used to reduce the probability of the rare event, by optimizing the model to produce a lower estimate. For example, this could be done using gradient descent, if the estimate were a differentiable function of the model's parameters. For an importance sampling method, this amounts to finding inputs for which the rare event occurs (i.e., red-teaming) and using them as training data, which is essentially the well-known method of adversarial training (Goodfellow et al., 2014). However, since our activation extrapolation methods do not correspond to any red-teaming method, new activation extrapolation methods potentially provide us with new ways to reduce the probabilities of rare events.

## 6.3 IMPORTANCE SAMPLING VERSUS ACTIVATION EXTRAPOLATION

In our experiments, we found that importance sampling methods outperformed activation extrapolation. Nevertheless, there are theoretical cases in which importance sampling performs worse than other methods. For example, consider a model that outputs the SHA-256 hash of its input: finding any input that gives rise to a particular output is computationally infeasible, yet it is still easy to estimate the probability of a particular output by modeling the output of the hash function as random.

More generally, we are excited about low probability estimation as a concrete problem for which for which it may be necessary to leverage internal model activations. In place of importance sampling, we may be able to use deductive estimates based on a presumption of independence (Christiano et al., 2022). Our Quadratic Logit Decomposition method is an early proof of concept of this, even though it is outperformed by importance sampling in our setting.

## 6.4 LIMITATIONS

There are two main limitations of our experimental setup. First, we only use input distributions that factor into independent tokens. This choice is necessary for the definition of ITGIS. It is also

very convenient for the implementation of MHIS, because it gives efficient sampling access to the proposal distribution. To move beyond independent token input distributions, we could define the input distribution to be the output of a separate generative model and adapt some of the current estimation methods appropriately.

Second, we only study model behaviors that consist of a single token sampled at temperature $0$. This is unrealistic because in practice, if we were concerned about specific single-token outputs, it would be easy to filter them out. In contrast, the types of behaviors we actually worry about likely involve long chains of autoregressive generation or interaction with the external world (e.g., when forming and executing a plan). We are excited to see future work extending our setting in this direction.

Nevertheless, it is worth noting that formally-defined distributions and behaviors are more general than they may initially seem. For example, we could formalize the event "$M$ writes buggy code", as: When $M$'s output is given to GPT-4 along with the prompt "Does this code contain any bugs? Let's think step by step.", does GPT-4 end its response with YES?

## 7 RELATED WORK

The problem of low probability estimation was previously considered in the context of computer vision by Webb et al. (2019), where they propose using an Adaptive Multi-Level Splitting algorithm with Metropolis Hastings. However, they only study the problem in the context of computer vision with continuous input spaces, and their approaches still require finding positive samples, unlike our activation extrapolation methods. Phuong et al. (2024) and Højmark et al. (2024) attempt to estimate the probability that a language model passes certain capability evaluations, even when its success rate is low, though their methods are not directly applicable to our formal setting.

Our importance sampling methods can be viewed as solving a special case of controlled text generation (Zhang et al., 2023) in which we want to sample from an autoregressive distribution conditioned on a property of the full output (in our case, that the last token is $t$). Yang & Klein (2021) do this by training Future Discriminators to steer model generation towards the desired attribute. Lew et al. (2023) approach the problem with a Sequential Monte Carlo steering approach; however, their in-filling algorithm doesn't provide any benefit over naive sampling when all tokens except the last are independent. These works don't consider the problem of low probability estimation.

Zhao et al. (2024) focus on the problem of estimating the partition function of an unnormalized target distribution over sequences, which is a more general case of our low probability estimation problem. Their Twisted Sequential Monte Carlo methods can be viewed as more advanced versions of our importance sampling methods. In contrast, in this work we focus on motivating the low probability estimation problem and introducing methods that do not involve searching for positive samples, such as activation extrapolation.

Finally, there is a large body of work applying adversarial training to improve worst-case model performance (Bai et al., 2021; Goodfellow et al., 2014; Ilyas et al., 2019), especially in the context of language models (Madry, 2017; Liu et al., 2020). Perez et al. (2022) explores using language models themselves to aid in red-teaming other models. Latent adversarial training (Casper et al., 2024; Sheshadri et al., 2024) generalizes standard adversarial training by optimizing over perturbations in activation space; this means that, like activation extrapolation methods, it can be effective even when the adversarial training search problem over input space is hard.

## 8 CONCLUSION

In this paper, we introduce the problem of low probability estimation along with four novel estimation methods. We define and collect ground-truth probabilities for $8$ different input distributions, then use them to evaluate the performance of our proposed methods. We find that the two importance sampling-based methods perform the best, with larger models favoring MHIS over ITGIS.

We are excited for future work that extends our empirical setup to non-independent input distributions and output behaviors that involve more than one token. We are also looking forward to future papers that develop more accurate estimation methods, especially methods like QLD that move beyond importance sampling.

## 9 ACKNOWLEDGEMENTS

Paul Christiano played a significant role in advising the project. We are grateful for intermediate theoretical contributions from David Matolcsi and George Robinson. Thanks additionally to Jean-Stanislas Denain and Eric Neyman for feedback on a draft.

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

# A  INPUT DISTRIBUTIONS

Table 2: Definitions of input distributions.

| Name | Tokens | Description |
|---|---|---|
| `hex` | 33 | Tokens that consist solely of hexadecimal characters, weighted by their frequency in long, uniformly-random hexadecimal strings. |
| `camel` | 33 | Tokens that start with a capital letter, then have only lowercase letters, weighted by their frequency in Python code. |
| `colon` | 34 | Tokens weighted by frequency in Python code. Always ends with a colon. |
| `if` | 34 | Tokens weighted by frequency in Python code. Always starts with '`if`'. |
| `caps` | 21 | Tokens that consist only of capital letters or punctuation, weighted by frequency in English text. Starts with '`He screamed:  "`' or '`She screamed:  "`'. |
| `english` | 26 | Tokens that consist only of letters and start with a space, as well as punctuation. Weighted by frequency in English text. |
| `spanish` | 25 | Tokens that consist only of letters and spaces. Weighted by frequency in Spanish text. |
| `icl` | 29 | A simple in-context learning prompt of the form '`A for _ R for _ C for _ ...`  `Y for _ ?  for`', where the underscores are replaced with random tokens that start with the corresponding letter (weighted by frequency in English text), and the `?` is replaced with a uniformly random letter. The letters spell out 'ARCTHEORY'. |

# B  FULL ESTIMATION ALGORITHMS

## I  ALGORITHM FOR ITGIS

The algorithm for ITGIS is as follows. Initialize $s_i^{(0)} \equiv 0$ for all $i$ and a step counter $j = 0$. Then, repeatedly:

1. Increase the step counter $j$ by 1.

2. Compute the distribution $q^{(j-1)}$ as defined by the previous score function $s_i^{(j-1)}$. This can be explicitly computed as a list of $|\mathcal{V}|$ probabilities per token position.

3. Draw a batch of samples from from $q^{(j-1)}$. Use the empirical mean of their gradient to estimate $\hat{s}_i^{(j)}(x) = \mathbb{E}_{\mathbf{x}' \sim q^{(j-1)}}[\nabla_{\mathbf{x}'} M_t(\mathbf{x}')]_{i,x}$ for all $i \in [k], x \in \mathcal{V}$.

4. Update the next $s_i^{(j)}$ for all $i$ according to an exponentially-weighted moving average:

$$s_i^{(j)} = \frac{\hat{s}_i^{(j)} + \alpha \hat{s}_i^{(j-1)} + \cdots + \alpha^{j-1} \hat{s}_i^{(1)}}{1 + \alpha + \alpha^2 + \cdots + \alpha^{j-1}}.$$

   In practice, we use $\alpha = 0.9$.

5. Calculate the average value of the importance sampling estimator $\frac{p(\mathbf{x})}{q^{(j-1)}(\mathbf{x})} \mathbb{1}[M(\mathbf{x}) = t]$ for all samples in this batch.

The final output of the method is the average value of this importance sampling estimator across all batches. The number of batches and samples per batch is specified in Appendix F. See Algorithm 1 for pseudocode.

---

**Algorithm 1** Independent Token Gradient Importance Sampling (ITGIS)

---

**Require:** Model $M$, target token $t$, input length $k$, token distributions $p_1, \ldots, p_k$, temperature $T$, iterations $n$, batch size $B$

1: $s_i^{(0)} \leftarrow \mathbf{0} \in \mathbb{R}^{|\mathcal{V}|}$ for all $i \in [k]$          # Initialize score functions
2: estimates $\leftarrow []$

3: **for** $j \leftarrow 1$ to $n$ **do**
4:     **for** $i \leftarrow 1$ to $k$ **do**
5:        $q_i^{(j-1)}(x) \leftarrow p_i(x) \cdot \exp(s_i^{(j-1)}(x)/T)$ for all $x \in \mathcal{V}$
6:        Normalize $q_i^{(j-1)}$ to have sum 1
7:     **end for**

8:     Sample $B$ inputs $\{\mathbf{x}^{(b)}\}_{b=1}^B$ from $q_1^{(j-1)} \times \cdots \times q_k^{(j-1)}$
9:     **for** $i \leftarrow 1$ to $k$ **do**
10:       $\hat{s}_i^{(j)}(x) \leftarrow \frac{1}{B} \sum_{b=1}^B [\nabla_{\mathbf{x}} M_t(\mathbf{x}^{(b)})]_{i,x}$ for all $x \in \mathcal{V}$
11:     **end for**

12:     $\alpha \leftarrow 0.9$
13:     **for** $i \leftarrow 1$ to $k$ **do**
14:       $s_i^{(j)} \leftarrow \frac{\hat{s}_i^{(j)} + \alpha \hat{s}_i^{(j-1)} + \cdots + \alpha^{j-1} \hat{s}_i^{(1)}}{1 + \alpha + \alpha^2 + \cdots + \alpha^{j-1}}$
15:     **end for**

16:     estimate $\leftarrow \frac{1}{B} \sum_{b=1}^B \frac{\prod_{i=1}^k p_i(\mathbf{x}_i^{(b)})}{\prod_{i=1}^k q_i^{(j-1)}(\mathbf{x}_i^{(b)})} \mathbb{1}[M(\mathbf{x}^{(b)}) = t]$
17:     Append estimate to estimates
18: **end for**

19: **return** $\frac{1}{n} \sum_{j=1}^n$ estimates$[j]$

---

## II  ALGORITHM FOR MHIS

We define the proposal distribution $\phi(\cdot|\mathbf{x})$ to be the distribution induced by the following procedure:

1.  Choose a random token position $i \in [k]$ to modify.

2.  Calculate the gradient at that token $[\nabla_{\mathbf{x}} M_t(\mathbf{x})]_i \in \mathbb{R}^{|\mathcal{V}|}$ (treating $\mathbf{x} = (\mathrm{x}_1, \ldots, \mathrm{x}_k)$ as a one-hot vector in $\mathbb{R}^{k \times |\mathcal{V}|}$). Call this gradient $\mathbf{g}$.

3.  Sample a replacement token $\mathrm{x}'_i$ from the distribution proportional to

$$p_i(x'_i) \cdot \exp\left(\frac{\mathbf{g}_{x'_i}}{T}\right).$$

4.  Output $\mathbf{x}' = (\mathrm{x}_1, \ldots, \mathrm{x}'_i, \ldots, \mathrm{x}_k)$.

Note that the transition probability in Metropolis–Hastings only depends on the ratio

$$\frac{q(\mathbf{x}')\phi(\mathbf{x}|\mathbf{x}')}{q(\mathbf{x})\phi(\mathbf{x}'|\mathbf{x})} = \frac{p_i(\mathrm{x}'_i)}{p_i(\mathrm{x}_i)} \cdot \exp\left(\frac{M_t(\mathbf{x}') - M_t(\mathbf{x})}{T}\right) \cdot \frac{\phi(\mathbf{x}|\mathbf{x}')}{\phi(\mathbf{x}'|\mathbf{x})},$$

which is easy to compute given forwards and backwards passes at $\mathbf{x}$ and $\mathbf{x}'$.

We use an initial burn-in period (see Appendix F) for the random walk before recording samples $\mathbf{x}^{(1)}, \ldots, \mathbf{x}^{(n)}$. The final output of the method is the empirical importance sampling estimate

$$\frac{1}{n} \sum_{j=1}^{n} \frac{p(\mathbf{x}^{(j)})}{q(\mathbf{x}^{(j)})} \mathbb{1}[M(\mathbf{x}^{(j)}) = t].$$

This requires computing $q(\mathbf{x})$, which involves the normalization constant. To save samples, we estimate the normalization constant using the identity:

$$\mathbb{E}_{\mathbf{x} \sim p}\left[\exp\left(\frac{M_t(\mathbf{x})}{T}\right)\right] = \mathbb{E}_{\mathbf{x} \sim q}\left[\exp\left(-\frac{M_t(\mathbf{x})}{T}\right)\right]^{-1}.$$

The right-hand side can be estimated using the $n$ samples we already have from (approximately) $q$. In practice, the way we estimate the normalizing constant does not matter much (most of the error comes from other steps). See Algorithm 2 for pseudocode.

---

**Algorithm 2** Metropolis–Hastings Importance Sampling (MHIS)

---

**Require:** Model $M$, target token $t$, input length $k$, token distributions $p_1, \ldots, p_k$, temperature $T$, number of burn-in steps $n_{\text{burn}}$, number of samples $n$
1: Initialize $\mathbf{x}$ by sampling from $p_1 \times \cdots \times p_k$
2: samples $\leftarrow$ []
3: **for** step $\leftarrow 1$ to $n_{\text{burn}} + n$ **do**
4:     $i \leftarrow \text{Unif}([k])$                                                    # Choose random position
5:     $\mathbf{g} \leftarrow [\nabla_{\mathbf{x}} M_t(\mathbf{x})]_i$
6:     Sample $\mathbf{x}'_i$ from $\propto p_i(x'_i) \cdot \exp(\mathbf{g}_{x'_i}/T)$                  # Proposed token
7:     $\mathbf{x}' \leftarrow (\mathbf{x}_1, \ldots, \mathbf{x}'_i, \ldots, \mathbf{x}_k)$                          # Proposed new state
        # Compute acceptance ratio $r = \frac{q(\mathbf{x}')\phi(\mathbf{x}|\mathbf{x}')}{q(\mathbf{x})\phi(\mathbf{x}'|\mathbf{x})}$
8:     $r \leftarrow \text{AcceptanceRatio}(M, t, T, (p_1, \ldots, p_k), \mathbf{x}, \mathbf{x}')$
9:     **if** $\text{Unif}(0, 1) < r$ **then**
10:       $\mathbf{x} \leftarrow \mathbf{x}'$                                                    # Accept proposal
11:     **end if**

12:     **if** step $> n_{\text{burn}}$ **then**
13:       Append $\mathbf{x}$ to samples
14:     **end if**
15: **end for**

16: $Z \leftarrow \left(\frac{1}{n} \sum_{j=1}^{n} \exp(-M_t(\text{samples}[j])/T)\right)^{-1}$         # Est. normalizing constant
17: **return** $\frac{1}{n} \sum_{j=1}^{n} \frac{\exp(M_t(\text{samples}[j])/T)}{Z} \mathbb{1}[M(\text{samples}[j]) = t]$

---

## III ALGORITHM FOR QLD

Recall that $\mathbf{v}(\mathbf{x}) \in \mathbb{R}^d$ is the random vector representing the activations of the model right before the unembedding step. After collecting $n$ samples of $\mathbf{v}^{(1)}, \ldots, \mathbf{v}^{(n)}$, we compute their empirical mean $\boldsymbol{\mu} \in \mathbb{R}^d$ and covariance $\boldsymbol{\Sigma} \in \mathbb{R}^{d \times d}$. Then define $\mathbf{u}$ to be the whitened version of $\mathbf{v}$:

$$\mathbf{u} := \boldsymbol{A}^{-1}(\mathbf{v} - \boldsymbol{\mu})$$
$$\mathbf{v} = \boldsymbol{A}\mathbf{u} + \boldsymbol{\mu},$$

where $\boldsymbol{A} \in \mathbb{R}^{d \times d}$ is any matrix such that $\boldsymbol{A}\boldsymbol{A}^\top = \boldsymbol{\Sigma}$. Note that $\mathbf{u}$ has mean 0 and covariance $\text{Id}_d$. From now on, we principally work in this whitened representation of activation space, as it has the convenient property that the $\mathbf{u} \cdot \boldsymbol{e}$ and $\mathbf{u} \cdot \boldsymbol{e}'$ are uncorrelated iff $\boldsymbol{e}$ and $\boldsymbol{e}'$ are orthogonal.

We choose the unit vector $\boldsymbol{d} \in \mathbb{R}^n$ to point in the direction of the shortest accepting vector (Appendix D), then decompose our whitened samples $\mathbf{u}^{(1)}, \ldots, \mathbf{u}^{(n)}$ into components parallel and perpendicular to $\boldsymbol{d}$:

$$\mathbf{a}^{(i)} := \boldsymbol{d}\boldsymbol{d}^\top \mathbf{u}^{(i)}$$
$$\mathbf{b}^{(i)} := \mathbf{u}^{(i)} - \mathbf{a}^{(i)}.$$

Finally, we output:

$$\frac{1}{n^2} \left| \left\{ (i, j) \in [n]^2 \,\middle|\, \mathbf{a}^{(i)} + \mathbf{b}^{(j)} \in S \right\} \right|,$$

where

$$S := \left\{ \boldsymbol{u} \in \mathbb{R}^d \,\middle|\, \arg\max_i((\boldsymbol{A}\boldsymbol{u} + \boldsymbol{\mu}) \cdot \boldsymbol{W}_U)_i = t \right\}.$$

This proportion can be computed in $\tilde{O}(n)$ time—we don't need to explicitly iterate over all $n^2$ pairs. By the convexity of the acceptance region $S$, for any fixed $\mathbf{b}$ there is a single interval $[\ell, r]$ such that $a \in [\ell, r] \Leftrightarrow a\boldsymbol{d} + \mathbf{b} \in S$. We can efficiently compute the bounds of this interval for every sample $\mathbf{b}^{(j)}$ by solving a linear system of inequalities, and then we can calculate how many $\mathbf{a}^{(i)}$ fall into each range in $O(\log n)$ time after sorting. Thus, the computational cost of QLD is dominated by running $n$ forwards passes of $M$ to generate the samples $\mathbf{u}^{(1)}, \ldots, \mathbf{u}^{(n)}$. See Algorithm 3 for pseudocode.

---

**Algorithm 3** Quadratic Logit Decomposition (QLD)

---

**Require:** Model $M$ with unembed matrix $\boldsymbol{W}_U$, target token $t$, sample size $n$
 1: Sample $n$ inputs $\mathbf{x}^{(1)}, \ldots, \mathbf{x}^{(n)}$ from input distribution
 2: Compute pre-unembed activations $\mathbf{v}^{(i)} \in \mathbb{R}^d$ by running $M$ on $\mathbf{x}^{(i)}$ for $i \in [n]$
 3: $\boldsymbol{\mu} \leftarrow \frac{1}{n} \sum_{i=1}^{n} \mathbf{v}^{(i)}$
 4: $\boldsymbol{\Sigma} \leftarrow \frac{1}{n} \sum_{i=1}^{n} (\mathbf{v}^{(i)} - \boldsymbol{\mu})(\mathbf{v}^{(i)} - \boldsymbol{\mu})^{\top}$
 5: Find $\boldsymbol{A}$ such that $\boldsymbol{A}\boldsymbol{A}^{\top} = \boldsymbol{\Sigma} + \epsilon \cdot \mathrm{Id}_d$       # via Cholesky decomposition
 6: **for** $i \leftarrow 1$ to $n$ **do**
 7:   $\mathbf{u}^{(i)} \leftarrow \boldsymbol{A}^{-1}(\mathbf{v}^{(i)} - \boldsymbol{\mu})$           # Whitened activations
 8: **end for**

 9: $\boldsymbol{d} \leftarrow \mathrm{ShortestAcceptingVector}(\boldsymbol{A}, \boldsymbol{\mu}, \boldsymbol{W}_U, t)$       # See Appendix D
10: **for** $i \leftarrow 1$ to $n$ **do**
11:   $\mathbf{a}^{(i)} \leftarrow \boldsymbol{d}\boldsymbol{d}^{\top}\mathbf{u}^{(i)}$           # Parallel component
12:   $\mathbf{b}^{(i)} \leftarrow \mathbf{u}^{(i)} - \mathbf{a}^{(i)}$          # Perpendicular component
13: **end for**

14: Sort $\{\mathbf{a}^{(i)}\}_{i=1}^{n}$ in ascending order
15: count $\leftarrow 0$
16: **for** $j \leftarrow 1$ to $n$ **do**
    # Solve linear inequalities to get $[\ell_j, r_j]$ such that $a \in [\ell_j, r_j] \Leftrightarrow a\boldsymbol{d} + \mathbf{b}^{(j)} \in S$
17:   $[\ell_j, r_j] \leftarrow \mathrm{FindAcceptanceInterval}(\mathbf{b}^{(j)}, \boldsymbol{d}, \boldsymbol{A}, \boldsymbol{\mu}, \boldsymbol{W}_U, t)$
18:   count $\leftarrow$ count $+ |\{i : \mathbf{a}^{(i)} \in [\ell_j, r_j]\}|$      # Binary search for bounds
19: **end for**

20: **return** count$/n^2$

---

## C    Principles for choosing a decomposition in QLD

In this section, we justify the claim that we rely on the following two assumptions of the decomposition $\mathbf{u} = \mathbf{a} + \mathbf{b}$ for QLD to perform well: 1) $\mathbf{a}$ and $\mathbf{b}$ are independent, and 2) the contribution towards the output behavior is split roughly equally between the two terms.

The first assumption is straightforward: if $\mathbf{a}$ and $\mathbf{b}$ are not independent, then $\mathbf{a} + \mathbf{b}'$ (where $\mathbf{b}'$ comes from an i.i.d. copy of $\mathbf{u}$) does not have the same distribution as $\mathbf{u}$. The failure of this assumption introduces bias into the estimation method. Working in whitened space ensures that $\mathbf{a}$ and $\mathbf{b}$ are uncorrelated, which is a first step towards independence.

The second assumption—that the contribution to the target behavior is roughly equally split between $\mathbf{a}$ and $\mathbf{b}$—is necessary for QLD to have an advantage over naive sampling. For purposes of illustration, say that $d = 2$ (so that both $\mathbf{a}$ and $\mathbf{b}$ can be treated as scalars), and that $\mathbf{a}, \mathbf{b} \overset{\text{iid}}{\sim} \mathcal{N}(0, 1)$. Then, consider three ways that the contribution to the target behavior could be split:

- *Scenario 1 (no split):* The target token is outputted iff $\mathbf{a} > 10$. In this case, QLD provides no advantage over naive sampling, as the proportion of $(\mathbf{a}^{(i)}, \mathbf{b}^{(j)})$ pairs in the acceptance region is exactly the same as the proportion of $(\mathbf{a}^{(i)}, \mathbf{b}^{(i)})$ pairs. If $p$ is the probability of the behavior, then $p^{-1}$ samples are required to consistently obtain a positive estimate.

- *Scenario 2 (even split):* The target token is outputted iff $\mathbf{a} + \mathbf{b} > 10\sqrt{2}$. In this case, QLD has a quadratic advantage over naive sampling. It only requires around $p^{-1/2}$ samples for QLD to consistently obtain a positive estimate.

- *Scenario 3 (uneven split):* The target token is outputted iff $\mathbf{a} + 2\mathbf{b} > 10\sqrt{5}$. The contribution here is split between $\mathbf{a}$ and $\mathbf{b}$, though not equally, so QLD's efficiency falls in between the previous two scenarios. We can calculate that it requires around $p^{-5/9}$ samples for QLD to consistently obtain a positive estimate.[7]

In practice, the condition for outputting the target token is more complex than a single linear constraint on $\mathbf{a}$ and $\mathbf{b}$. Nevertheless, these examples motivate the idea that the more evenly we can split contribution between two subspaces, the lower variance our estimator will have.

Given that $\mathbf{b}$ has $d - 1$ dimensions while $\mathbf{a}$ only has 1, most choices of $\boldsymbol{d}$ will end up giving $\mathbf{b}$ much more influence over the behavior than $\mathbf{a}$. This motivates us to identify a particularly important direction with $\boldsymbol{d}$; in some informal sense we want to find a direction that is "$d/2$ times more important" than the average direction.

When we run QLD with many more samples than its standard budget of $2^{16}$, its performance improves but plateaus at a level that is still worse than the sampling methods. This shows that QLD is currently limited by a poor independence assumption, as the error arising from an unequal split of contribution should vanish with a sufficient number of samples.

---

[7]In general, if the condition is $\alpha\mathbf{a} + \sqrt{1 - \alpha^2}\mathbf{b} > 10$, it requires roughly $p^{-1/\left(\alpha + \sqrt{1-\alpha^2}\right)^2}$ samples.

# D  COMPUTING THE SHORTEST ACCEPTING VECTOR

Recall that the acceptance region $S \subseteq \mathbb{R}^d$ is the subset of whitened pre-unembed space that results in the model outputting $t$:

$$S := \left\{ \boldsymbol{u} \in \mathbb{R}^d \;\middle|\; \arg\max_i((\boldsymbol{A}\boldsymbol{u} + \boldsymbol{\mu}) \cdot \boldsymbol{W}_U)_i = t \right\}.$$

We define $\boldsymbol{d}$ to point in the direction of the shortest vector in $S$, i.e., $\arg\min_{\boldsymbol{u} \in S} \|\boldsymbol{u}\|$. This vector can be approximated using an iterative convex projection algorithm.

$S \subseteq \mathbb{R}^d$ is an intersection of $|\mathcal{V} - 1|$ half-spaces $H_1, \ldots, H_{t-1}, H_{t+1}, \ldots H_{\mathcal{V}}$, where $H_i$ represents the set of all activations (in whitened pre-unembed space) that result in the logit on token $t$ being larger than the logit on token $i$.

Given any convex set $C$ (such as a half-plane), the *projection* of $\boldsymbol{x} \notin C$ onto $C$ is $\arg\min_{\boldsymbol{x}' \in C} \|\boldsymbol{x} - \boldsymbol{x}'\|_2$. Given a collection of convex sets, there exists a simple algorithm for finding a point in their intersection: start with an arbitrary point $\boldsymbol{x}$, then repeatedly project $\boldsymbol{x}$ onto a random convex set that does not already contain $\boldsymbol{x}$. Eventually, this process converges to a point in their intersection (Gubin et al., 1967).

We apply this method to find an element of $S$. To ensure that it is the shortest element, we also project $\boldsymbol{x}$ onto balls centered at 0 with smaller and smaller radii by multiplying $\boldsymbol{x}$ by 0.99. The exact procedure is described in Algorithm 4.[8] In practice, it always takes much less than $100 \cdot n_{\text{reps}}$ steps for the algorithm to return a value.

---

**Algorithm 4** Random Constraint Projection

---

**Require:** Half-spaces $H_1, \ldots, H_{t-1}, H_{t+1}, \ldots, H_{|\mathcal{V}|}$, number of repetitions $n_{\text{reps}}$
1:  $\boldsymbol{x} \leftarrow \boldsymbol{0} \in \mathbb{R}^d$
2:  **for** step_cnt $\leftarrow 1$ to $100 \cdot n_{\text{reps}}$ **do**
3:      Pick a random $i$ among all $i$ such that $\mathbf{x} \notin H_i$
4:      Project $\boldsymbol{x}$ onto $H_i$
5:      **if** $\boldsymbol{x}$ lies in $S$ (up to some tolerance $\epsilon$) **then**
6:          **if** step_cnt $< n_{\text{reps}}$ **then**
7:              Scale $\boldsymbol{x}$ by 0.99.
8:          **else**
9:              **return** $\boldsymbol{x}$
10:         **end if**
11:     **end if**
12: **end for**

---

[8]We found a few minor bugs in our implementation of the $\epsilon$-tolerance in our algorithm after we ran experiments, but we don't expect them to have affected the results at all.

# E GROUND TRUTH TOKEN DISTRIBUTION

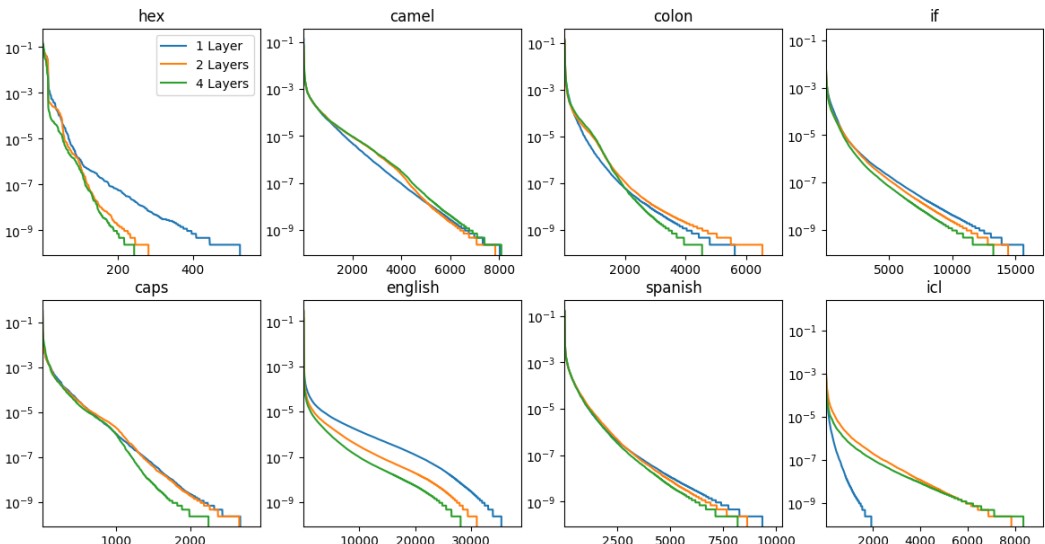

Figure 5: The ground truth probabilities of tokens for each distribution and model size, sorted from most to least probable (the height of the curve at position $x$ is the probability of the $x$-th most common token). Any tokens that appeared $0$ times across all $2^{32}$ samples are not plotted. The hex distribution only had $159$ and $135$ tokens in the range $[10^{-9}, 10^{-5}]$ for the 2- and 4-layer models, respectively, so we used every such token instead of sampling 256 of them.

## F    COMPUTATIONAL BUDGETS

Each estimation method was given a budget of roughly $2^{16}$ model calls. More specifically:

- Independent Token Gradient Importance Sampling uses $2^8$ batches of size $2^8$, for a total of $2^{16}$ samples. The average gradient is updated after each batch. Note that this method requires backwards passes as well as forwards passes.

- Metropolis–Hastings Importance Sampling uses $2^{10} + 2^{11}$ batches of size $2^5$, for a total of $1.5 \cdot 2^{16}$ samples (the batch size indicates the number of independent random walks the method simulates). The first $2^{10}$ batches are used as a burn-in period for the random walk and are discarded, so only $2^{16}$ samples are actually used to calculate the estimate.

- Quadratic Logit Decomposition uses $n = 2^{16}$ samples of the pre-unembed activation $\mathbf{v}$. The MLE direction is approximated with $n_{\text{reps}} = 200$ iterations of the Random Constraint Projection algorithm (Appendix $D$); this makes up a trivial fraction of the total compute usage of the method).

- Gaussian Logit Difference uses $2^{16}$ samples of the logit difference to estimate $\mu$ and $\sigma$, the mean and standard deviation of the difference between the target logit and the maximum logit. Note that in practice, the $\mu$ and $\sigma$ can be accurately estimated with much fewer than $2^{16}$ samples.

In practice, ITGIS and MHIS take the longest to test because they require separate samples for each target token. In contrast, QLD reuses the same $2^{16}$ samples of $v$ for all 256 target tokens associated with a given behavior.

# G  ALL METHOD PERFORMANCES

Table 3 shows the Itakura–Saito loss $(p/q - \ln(p/q) - 1)$ of all estimation methods on all input distributions and model sizes.

Table 3: Itakura–Saito loss comparison of all methods, distributions, and model sizes.

(a) 1-layer model

| Distribution | Constant | GLD | QLD | ITGIS | MHIS |
|---|---|---|---|---|---|
| hex | 2.5891 | 2.2163 | 2.0038 | 2.0484 | **1.3960** |
| camel | 2.5908 | 2.4419 | 2.0648 | **1.1997** | 2.0187 |
| colon | 2.7770 | 2.7091 | 1.2786 | 1.2209 | **1.0267** |
| if | 2.2424 | 2.1872 | 1.2321 | **1.0916** | 1.4455 |
| caps | 2.6619 | 2.6147 | 1.9413 | **1.4788** | 2.4023 |
| english | 1.9095 | 1.8120 | 1.2409 | 1.4539 | **0.7017** |
| spanish | 2.6079 | 2.4463 | **1.5628** | 1.6396 | 2.1538 |
| icl | 2.5467 | 2.4328 | 2.2373 | **0.7992** | 1.2344 |
| Average | 2.4906 | 2.3575 | 1.6952 | **1.3665** | 1.5474 |

(b) 2-layer model

| Distribution | Constant | GLD | QLD | ITGIS | MHIS |
|---|---|---|---|---|---|
| hex | 2.8839 | 2.8453 | 2.7698 | 2.6652 | **1.4985** |
| camel | 2.3242 | 2.2268 | 2.2679 | **1.9221** | 2.0637 |
| colon | 2.9893 | 2.8335 | 2.1215 | 2.2986 | **1.9042** |
| if | 2.6172 | 2.4377 | 1.8397 | **1.0301** | 1.2516 |
| caps | 2.6345 | 2.6820 | 2.4847 | 1.9271 | **1.6981** |
| english | 2.0989 | 2.0956 | 1.3462 | **0.9908** | 1.4480 |
| spanish | 2.5442 | 2.3662 | 1.5670 | **1.0951** | 2.3095 |
| icl | 2.6381 | 2.5419 | 2.3601 | 1.6420 | **1.0502** |
| Average | 2.5913 | 2.5036 | 2.0946 | 1.6964 | **1.6530** |

(c) 4-layer model

| Distribution | Constant | GLD | QLD | ITGIS | MHIS |
|---|---|---|---|---|---|
| hex | 2.4803 | 2.2934 | 2.4282 | 2.3090 | **2.0830** |
| camel | 2.4895 | 2.3271 | 2.4534 | **2.0937** | 2.2961 |
| colon | 2.9325 | 2.7319 | 2.4382 | 2.1295 | **1.1710** |
| if | 2.6477 | 2.5408 | 1.8313 | 1.8430 | **0.9261** |
| caps | 2.5970 | 2.5382 | 2.4142 | 2.0484 | **1.2972** |
| english | 2.7008 | 2.7432 | 1.5681 | **0.9943** | 1.3051 |
| spanish | 2.6415 | 2.5022 | 1.7716 | **1.6611** | 2.1936 |
| icl | 2.5029 | 2.1925 | 2.3866 | 2.2971 | **0.5477** |
| Average | 2.6240 | 2.4837 | 2.1615 | 1.9220 | **1.4775** |

## H    SQUARED ERROR IN LOG-SPACE

Figure 6 and Table 4 show the method performances when measured using squared error in log-space loss (i.e., $(\log p - \log q)^2$) instead of Itakura–Saito loss. The results are qualitatively identical using either metric. Note that we use separate affine fits to minimize each loss function—in Table 4 we naturally report the results of the fit corresponding to squared error in log-space. However, the importance sampling temperatures are not changed between the two metrics (they were tuned while minimizing Itakura–Saito loss).

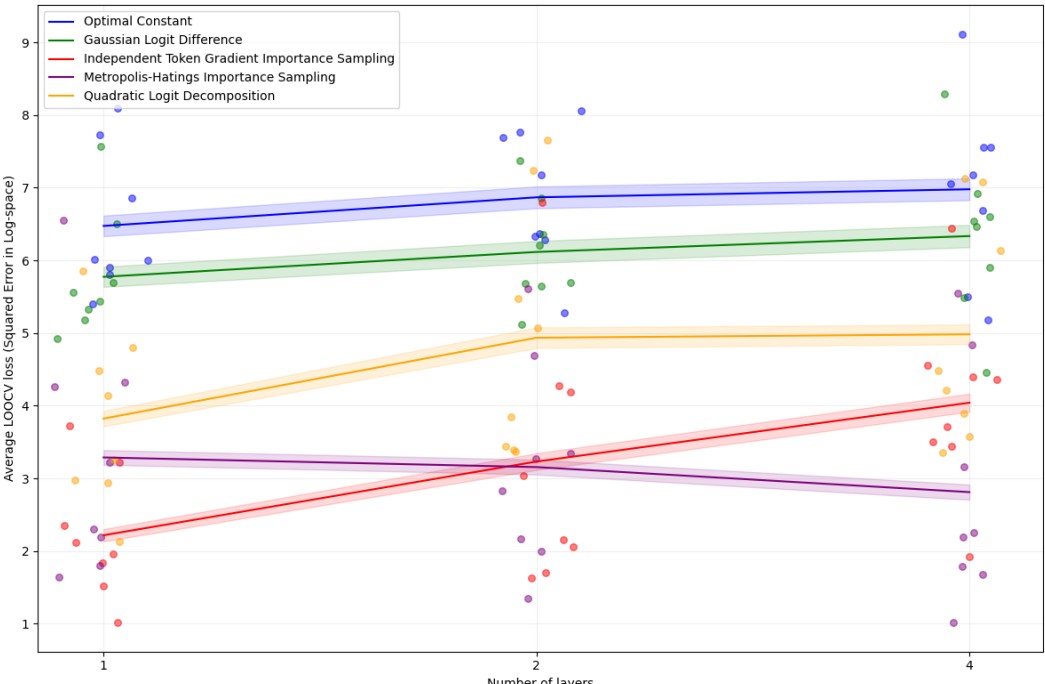

Figure 6: The squared error in log-space loss of all methods across different model sizes. The solid lines indicate the loss of each method averaged over all 8 distributions, with bands indicating standard error. The colored points indicate the loss on individual distributions, with horizontal jitter added for visibility.

Table 4: Squared error in log-space loss comparison of all methods, distributions, and model sizes.

(a) 1-layer model

| Distribution | Constant | GLD | QLD | ITGIS | MHIS |
|---|---|---|---|---|---|
| hex | 6.0093 | 5.1815 | 4.4863 | 3.7179 | **2.1963** |
| camel | 7.7295 | 6.4975 | 5.8488 | **1.5156** | 4.3248 |
| colon | 6.8554 | 5.4361 | 2.1379 | 2.3549 | **1.8061** |
| if | 5.3977 | 4.9287 | 2.9356 | **1.9555** | 3.2223 |
| caps | 8.0927 | 7.5675 | 4.1349 | **3.2173** | 6.5550 |
| english | 6.0051 | 5.3220 | 3.2639 | 1.8357 | **1.6395** |
| spanish | 5.7990 | 5.5649 | 2.9740 | **2.1231** | 4.2612 |
| icl | 5.9035 | 5.6915 | 4.8023 | **1.0222** | 2.3030 |
| Average | 6.4740 | 5.7737 | 3.8230 | **2.2178** | 3.2885 |

(b) 2-layer model

| Distribution | Constant | GLD | QLD | ITGIS | MHIS |
|---|---|---|---|---|---|
| hex | 8.0536 | 7.3674 | 7.6533 | 6.8004 | **1.9913** |
| camel | 7.6849 | 6.3518 | 7.2325 | **4.2695** | 5.6046 |
| colon | 7.1728 | 5.6416 | 3.3924 | 4.1851 | **2.8242** |
| if | 6.3346 | 5.6923 | 3.3695 | **1.7059** | 2.1639 |
| caps | 7.7682 | 6.8605 | 5.4771 | **3.0411** | 3.3432 |
| english | 5.2742 | 5.1185 | 3.4466 | **1.6335** | 3.2728 |
| spanish | 6.3718 | 5.6877 | 3.8404 | **2.1582** | 4.6937 |
| icl | 6.2785 | 6.2061 | 5.0740 | 2.0615 | **1.3515** |
| Average | 6.8673 | 6.1158 | 4.9357 | 3.2319 | **3.1556** |

(c) 4-layer model

| Distribution | Constant | GLD | QLD | ITGIS | MHIS |
|---|---|---|---|---|---|
| hex | 7.5559 | 6.9208 | 7.0750 | 6.4364 | **3.1605** |
| camel | 7.5597 | 6.5413 | 7.1263 | **3.5067** | 5.5501 |
| colon | 7.1791 | 6.6035 | 3.5817 | 4.3916 | **1.7897** |
| if | 6.6900 | 5.9072 | 3.8987 | 3.7153 | **1.6824** |
| caps | 9.1105 | 8.2957 | 6.1383 | 4.3615 | **2.2514** |
| english | 5.5003 | 5.4866 | 3.3561 | **1.9180** | 2.1938 |
| spanish | 7.0490 | 6.4641 | 4.2083 | **3.4425** | 4.8425 |
| icl | 5.1793 | 4.4569 | 4.4772 | 4.5584 | **1.0216** |
| Average | 6.9780 | 6.3345 | 4.9827 | 4.0413 | **2.8115** |

## I  TEMPERATURE TUNING

Both importance sampling methods require choosing a temperature parameter $T$. To tune $T$, we sweep over 9 different temperatures from $0.2$ to $5$, uniformly spaced in log-space. We choose the value of $T$ that achieves the lowest loss on 100 randomly chosen tokens with ground-truth probabilities in the range $[10^{-5}, 10^{-3}]$ to prevent over-fitting. We tune separate temperatures for each distribution, model size, and importance sampling method, shown in Table 5. It is likely that spending more effort to tune these temperatures (e.g., by tuning on more and rarer tokens) would moderately improve the final performances of the importance sampling methods.

Table 5: Temperatures $T$ used for the different methods.

| Distribution | 1 layer ITGIS | 1 layer MHIS | 2 layers ITGIS | 2 layers MHIS | 4 layers ITGIS | 4 layers MHIS |
|---|---|---|---|---|---|---|
| hex     | 1.00 | 0.67 | 1.50 | 0.67 | 5.00 | 0.67 |
| camel   | 1.00 | 2.24 | 1.50 | 2.24 | 1.00 | 2.24 |
| colon   | 1.00 | 1.00 | 1.00 | 1.50 | 0.67 | 1.00 |
| if      | 1.00 | 2.24 | 0.45 | 1.50 | 1.00 | 1.00 |
| caps    | 1.50 | 3.34 | 0.45 | 1.50 | 0.67 | 1.00 |
| english | 0.45 | 1.50 | 0.67 | 2.24 | 0.45 | 1.50 |
| spanish | 0.67 | 2.24 | 0.67 | 2.24 | 1.00 | 2.24 |
| icl     | 0.45 | 1.00 | 0.30 | 0.67 | 3.34 | 0.67 |

## J    PLOTS OF METHOD OUTPUTS

Figures 7, 8, and 9 show the outputs of the four methods on all three model sizes, using log-log plots of ground-truth probability vs method output. All graphs show the outputs after the Itakura–Saito fit has been applied (see Section 4.2). The horizontal lines of points reveal the value of the additive constant in the fit; any outputs of 0 will all lie on this line after the fit is applied.

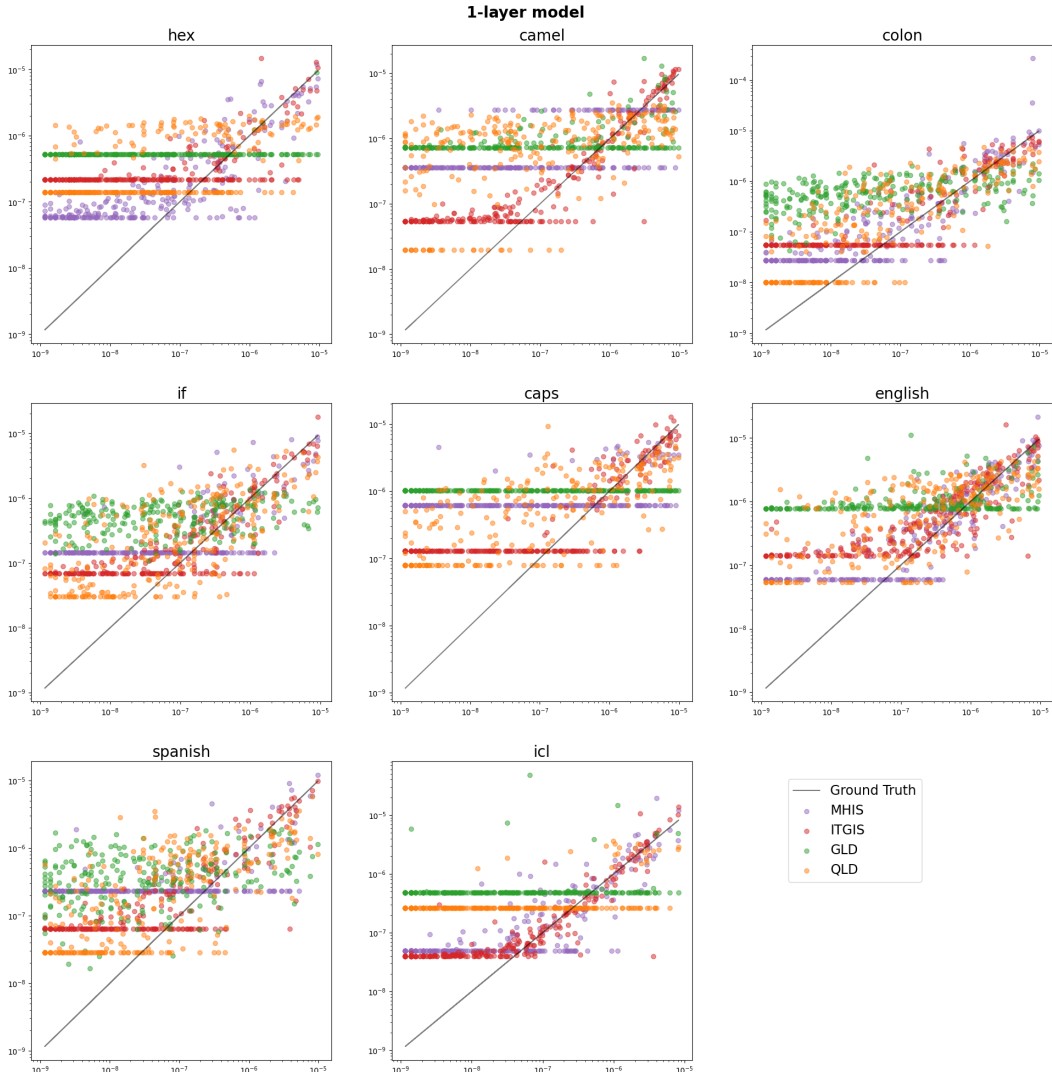

Figure 7: The outputs of methods, after a fit is applied, on all 256 tokens for each distribution on the 1-layer model. The horizontal axis represents the ground-truth token probability, while the vertical axis is the output of the model after a fit.

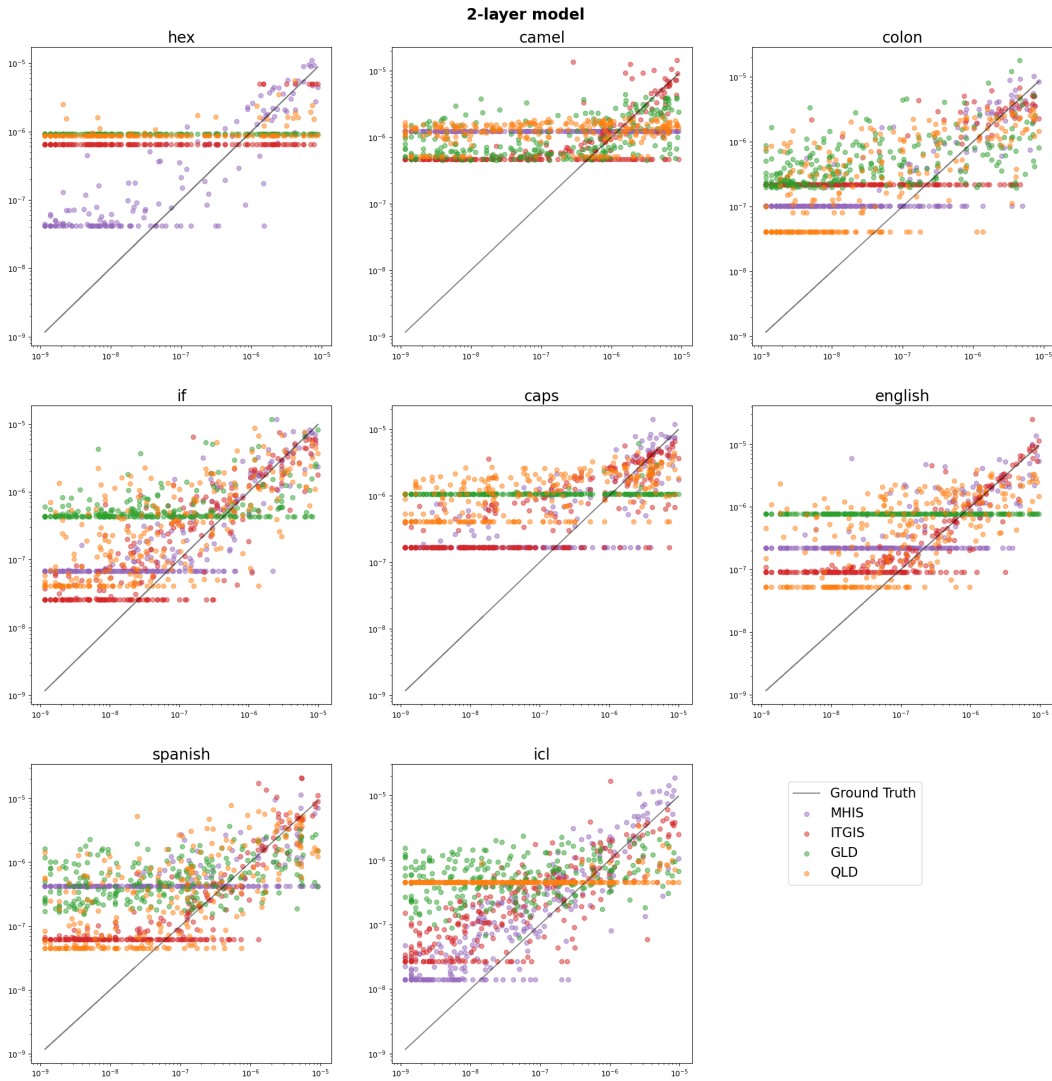

Figure 8: The outputs of methods, after a fit is applied, on all 256 tokens for each distribution on the 2-layer model. The horizontal axis represents the ground-truth token probability, while the vertical axis is the output of the model after a fit.

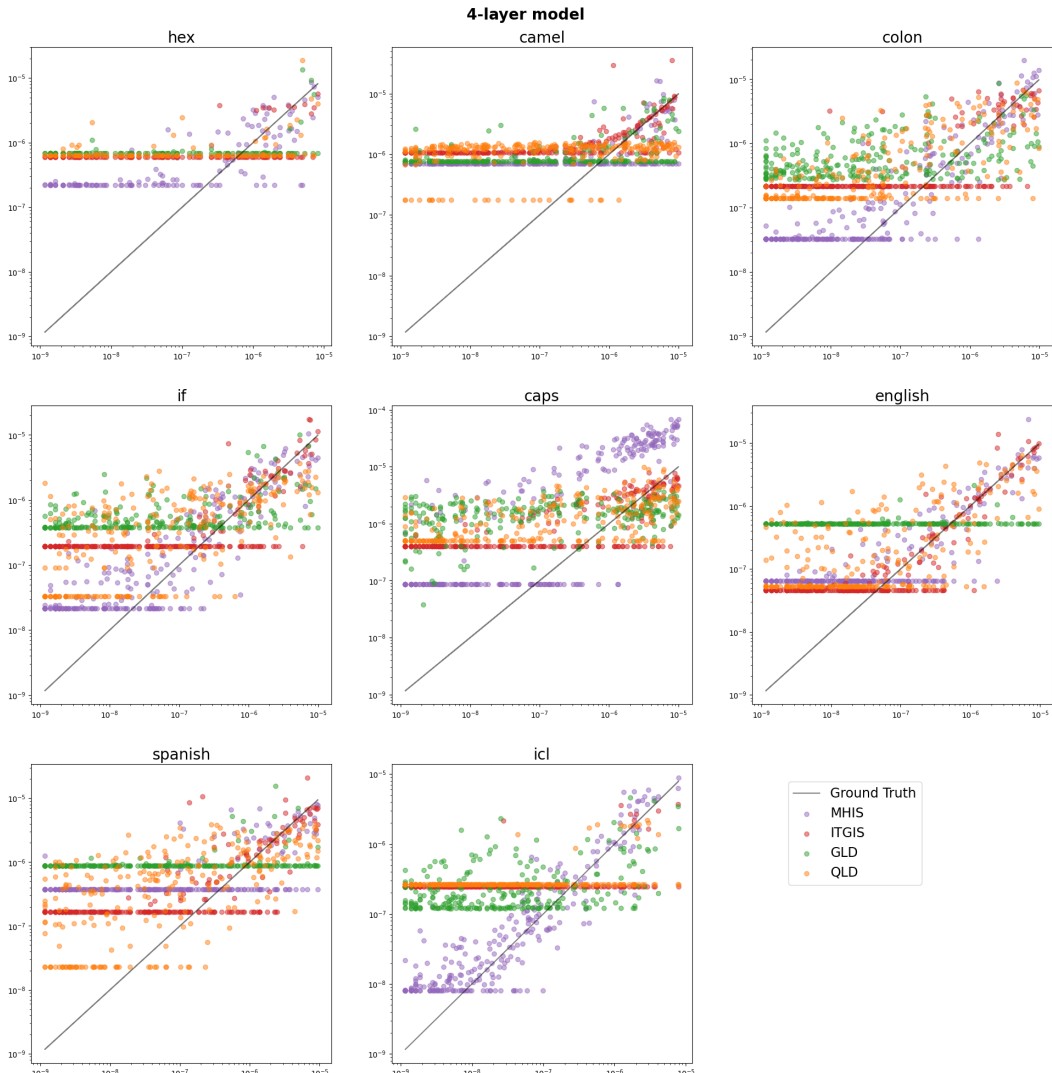

Figure 9: The outputs of methods, after a fit is applied, on all 256 tokens for each distribution on the 4-layer model. The horizontal axis represents the ground-truth token probability, while the vertical axis is the output of the model after a fit.

