# OpenReview forum: "Estimating the Probabilities of Rare Outputs in Language Models"
_ICLR.cc/2025/Conference — ICLR 2025 Spotlight_

### Official Review · Reviewer_sH4y · 2024-10-17

**Soundness:** 3
**Presentation:** 2
**Contribution:** 3
**Rating:** 8
**Confidence:** 3

**Summary:**

The paper studies the problem of estimating the probabilities of rare outputs of transformer models. The authors explore two ways to estimate these probabilities: i) importance sampling and ii) activation extrapolation, which is more novel, but less effective. For each method, they propose two sub-strategies. Results show that importance sampling performs best at estimating the probability of rare outputs and both proposed methods perform better than random sampling.

**Strengths:**

The paper focuses on an important and, in my opinion, understudied problem of deep learning in general, which is estimating the probability of rare, but potentially catastrophic outputs in deep learning models, and by so doing also capturing the distribution of inputs that would cause them. As stated by the authors correctly, solving this problem is key in advancing adversarial training and even reinforcement learning in many settings.

The approaches tested are principled and thoroughly derived and, while limited in the current applicability (see weaknesses), provide a starting point for further studies and developments of new methods to quantify the probability of rare outputs and all its applications.

**Weaknesses:**

Two main weaknesses:

1) The methods provided are quite limited in their application to larger models and real-world scenarios, and not just because the authors chose to test with small models, but also because the methods themselves perform computations that restrict their use. Specific points that introduce restrictions to be clarified:
a. The proposed method can only work with target behaviours defined as single token generation, i.e., define specific tokens that are considered rare for the next generation. This is quite limiting, as in language models target/dangerous behaviours are more often associated with semantic meaning of entire sentences and not with a single next token prediction, e.g., alignment for harmfulness.
b. Both methods involve computing the gradient of the rare token generation probability wrt the input tokens/sequence. This reinforces point a. above and further necessitates to have full access to the model, as we need to be able to compute this gradient and propagate it through the model, which is not trivial if the target behaviour is not simply a single specific token or the model is black-box.

2) The technical section describing the two method could be clearer. I suggest converting the point-by-point methods descriptions of supplementary B into pseudo-codes to make it clear what are the exact procedures to implement the different proposed methods and, if possible, put some of them in the main text. In the current form, the methods are given little space and I could not follow them without carefully reading supplementary B.

**Questions:**

In short, I believe this paper to be a good contribution to the community, providing a starting point to address the problem of rare output probability estimation. However, referencing the weaknesses above, I advise to address the following two points:

1) State more clearly that proposed methods are restricted in their applicability to different models and more complex scenario, in particular wrt the need to compute and propagate the gradient of the output probability.

2) detail more clearly the proposed strategies, ideally with pseudo-codes.

---

> ### Author Response · Authors · 2024-11-20
> **Response to Reviewer sH4y**
>
> We thank the reviewer for their detailed feedback. We share the reviewer’s opinion that addressing the possibility of rare but harmful model outputs is a key problem in advancing adversarial robustness and reinforcement learning.
>
> a: Regarding the reviewer’s first point about the limitations of behaviors defined by a single token, we acknowledge that this is one of the two major limitations of our current setup (see Section 6.4). However, we’d also like to point out that single-token behaviors can be somewhat expressive. For example, you could measure the probability that a language model misclassifies offensive text as benign by specifying an input distribution of offensive text (surrounded by the prompt “Is this text harmful or benign? [TEXT] Answer:”), then estimating the probability that the model outputs "BENIGN". Of course, this still fails to capture more behaviors that may arise over the course of multiple forward passes of chain-of-thought.
>
> b: Regarding the reviewer’s second point about our methods requiring full model access: While a black-box setting may be interesting to consider, we are primarily focused on the setting in which a model developer applies low probability estimation (as a complement to adversarial training) to prevent harmful model outputs; in this case, the developer should have full (white-box) access to their own model. None of these methods (including standard adversarial training) would be useful in this regard unless the model can be appropriately updated or secured, which presupposes full access. One of our primary motivations for this work is the intuition that model internals can be leveraged to improve model robustness beyond what can be done with black-box techniques.
>
> (Minor, but we also point out that our activation extrapolation methods don’t require gradients, and only make use of samples of the final-layer model activations. For these methods, the entire model up until the unembedding layer can be black-boxed.)
>
> Finally, we have accepted the reviewer’s suggestion to convert the full algorithms into pseudocode, which are now included in Appendix B. Unfortunately, we lack space to promote this pseudocode into the main body of the paper.

---

> ### Comment · Reviewer_sH4y · 2024-11-27
> **Response to Clarifications**
>
> I thank the reviewers for their clarifications.
>
> a. I can see that this limitation is clearly stated in the limitations section and I apologise for having missed it. I would still suggest to make it somewhat clear in the introduction. This work does not solve the problem of estimating rare/harmful outputs probability for most settings of practical relevance, but it is a good starting point to then develop more advanced solutions that could scale to these settings. If possible, I would also suggest ways to extend this to multi-token outputs and semantic meanings.
>
> b. I agree that success in this task even only for accessible models is still useful. I still suggest adding this as a limitation. I disagree about the fact that it would not be useful for black-box models. Many systems currently rely on proprietary API endpoints for a range of applications, e.g., customer service providers using openAI endpoints. As a service provider consuming a black-box model, knowing how likely it is to, e.g., be rude to customers, would be very helpful to chose which LLM provider and particular model to use.
>
> Given the Authors clarifications, I Raise my score.

---

### Official Review · Reviewer_fKZ4 · 2024-11-03

**Soundness:** 3
**Presentation:** 3
**Contribution:** 2
**Rating:** 5
**Confidence:** 4

**Summary:**

This work focus on probability estimation of low probability outputs in LLMs with argmax decoding. The paper explores the use of importance sampling and activation extrapolation for better estimate of these rare output events. The paper also provides experiments showing the relative performance of the various proposed algorithms.

**Strengths:**

- The paper is well written.
- The paper provides simple methods for low probability estimation in LLMs.
- Empirical results validate the working of their proposed methods (the importance sampling works better as shown by the authors).

**Weaknesses:**

- One major drawback I find is the lack of clear goal of the paper. The paper seems to provide some interesting algorithm for low probability estimation and high level discussion of how they can be used in practice. But there is no clear application or experiment that shows the usefulness of the contribution. I would urge the authors to kindly provide some clear applications and experiments to show its usefulness.

**Questions:**

Please see the weaknesses.

---

> ### Author Response · Authors · 2024-11-20
> **Response to Reviewer fKZ4**
>
> We thank the reviewer for their criticism.
>
> We discuss applications of low probability estimation in the Discussion section. Namely, we are interested in using low probability estimation as an alternative to standard adversarial training to limit rare but unacceptable model outputs, for example by directly training against the output of a LPE method to drive that probability down. Our response to Reviewer D4d5 goes into more details about how this could be accomplished.
>
> We agree with the reviewer that, at the moment, the activation extrapolation methods are not strong enough to offer immediate applications that outperform what can already be done with importance sampling methods. However, we think this paper makes an important first step by 1) motivating and defining the problem of low probability estimation as an alternative to adversarial training, and 2) introducing activation extrapolation as a novel class of methods. Our empirical results demonstrate that activation extrapolation methods handily beat baselines.
>
> If the reviewer feels that our elaboration has mitigated their concerns, we respectfully ask that they increase their score.

---

> > ### Comment · Reviewer_fKZ4 · 2024-11-24
> >
> > I thank the authors for their response. I went through all the reviews and their responses.
> >
> > My concern about the goal of the paper still remains. As also described by reviewer D4d5, the importance sampling method for LPE simply reduces to adversarial training, hence, does not add any helpful new method of training. On the other hand, the activation extrapolation method does provide a novel method but its performance is poor compared to importance sampling.
> >
> > The authors also claim that better methods may appear for LPEs in the future (in response to reviewer D4d5), which could help improve the worst-case behavior of models, but, I don't think this justifies the limited usefulness of the proposed methods in this paper.
> >
> > I think the paper does have a good motivation and even the activation extrapolation method seems interesting and novel, but more effort is needed to make that direction more useful than naive importance sampling/adversarial training.
> >
> > Despite the drawbacks, I am happy to increase the score for the potential future usefulness of the activation extrapolation direction as discussed by the authors in response to reviewer D4d5. But overall, I feel this paper needs more work to make the methods useful to the research community.

---

### Official Review · Reviewer_D4d5 · 2024-11-03

**Soundness:** 3
**Presentation:** 4
**Contribution:** 3
**Rating:** 6
**Confidence:** 2

**Summary:**

This paper introduces methods for estimating the probability of sampling certain outputs when conditioned on an input distribution in cases where naive sampling is infeasible. Specifically, they introduce four methods (two based on importance sampling and two based on extrapolating model activations) for the special case where the input distribution is independent per token and the output is a single token.

**Strengths:**

I am not familiar with prior work in this domain, but it appears as if the paper introduces novel methods for trying to estimate the probabilities of certain outputs given an input distribution.

Many intuitions are described in prose and are very clear/helpful:
- Why MHIS may be needed, why this is better for higher capacity models
- Why activation extrapolation methods could be better in certain scenarios.
- Why formally defined distribution can be relevant.

The experiments are sound and the results show that the proposed methods are clearly better than the naive sampling baseline.

Graphics are generally clear!

**Weaknesses:**

To my understanding, the main motivation described for this problem stems from using these methods to reduce the probability of rare but undesirable outputs. Although a reasonable motivation, there are limited results in this direction because, as the authors point out, doing this for importance sampling reduces to adversarial training, and the activation extrapolation methods are currently worse than the importance sampling methods.
- A more thoroughly description of how you would do this for activation extrapolation methods would be good.

The current problem as formulated is restricted (ie. single token output detection) and not practically useful. However the authors recognize this, and I agree that it is a good first step.

The description of the activation extrapolation methods in the main text could be made more clear, for example the motivation for why the QLD directions could be summarized in the main text (without this, it is hard to understand the method).

Nits:
- Add lines of best fit for figure 1 and 3.

**Questions:**

Is there any concrete benefit that the proposed methods add to the overall motivation of minimizing adversarial outputs (ex. based on the activation extrapolation)?

---

> ### Author Response · Authors · 2024-11-20
> **Response to Reviewer D4d5**
>
> We thank the reviewer for their detailed comments on both the strengths and weaknesses of the paper.
>
> We agree with the reviewer that the current state of activation extrapolation methods do not immediately offer a new method for reducing the probability of rare but undesirable outputs, as they currently underperform standard importance sampling. However, if future work identifies more successful LPE methods (whether through activation extrapolation or some other technique), we believe they may be useful for preventing worst-case model behaviors in the following ways:
> - As stated in the paper, if the low probability estimate is a differentiable function of the model's parameters, we could apply gradient descent against this estimate. For example, with Quadratic Logit Decomposition we could draw a batch of $n$ activation samples, mix-and-match them to generate $n^2$ synthetic samples, then backprop against a loss function defined on these $n^2$ synthetic samples (note that for a fixed draw of inputs, each synthetic sample is differentiable w.r.t. the model’s parameters). We do not expect this particular algorithm to work well in practice because the independence assumption made by QLD is poor.
> - We could use LPE to get a sense of worst-case model behaviors even if we don’t train against it. For example, a safety-conscious AI developer could apply a LPE method to each new model they train, and commit to taking special precautions if it returns a > 1e-20 chance of executing a catastrophic action on a random input. These “special precautions” could include: not deploying the model, auditing the training process, alerting regulators, or retraining the model with a different random seed.
>
> More broadly, we note that current adversarial training methods rarely make use of model internals (besides gradient access), and we think there is value in exploring techniques that take advantage of our ability to understand or manipulate internal activations, such as activation extrapolation. One other good example of this is Latent Adversarial Training, which we added to our Related Works.
>
> Regarding the clarity of the description of QLD, we have added a small clarification on how the two principles relate to the bias and variance of the estimator. Unfortunately, the pagecount limit prevents us from moving more from the appendix into the main body. We also have added pseudocode for all of the methods in the appendix.
>
> Finally, regarding the nit: we worry that lines of best fit would be misleading on Figures 1 and 3 because many of the points are at $y=-\infty$. Additionally, the affine fits we use are of the form $ax^c + b$, which don’t necessarily correspond to lines on a log-log plot (unless $b = 0$). Thus, having a line of best fit that is close to the ground truth diagonal $y=x$ wouldn’t tell us much about the overall performance of that method.

---

### Official Review · Reviewer_DbYe · 2024-11-04

**Soundness:** 3
**Presentation:** 4
**Contribution:** 4
**Rating:** 10
**Confidence:** 4

**Summary:**

This paper studies low-probability estimation - the problem of estimating the probability that model outputs will satisfy a certain criteria on inputs sampled from some distribution. The authors are primarily concerned with models/behaviors where the probability is hard to estimate empirically using unbiased samples from the input distribution, due to the event being rare.

More specifically, the authors study the task of estimating the percentage of inputs on which a transformer model predicts a certain token $t$ as being the most likely next token. Three methods are tested - two importance sampling methods, which leverage gradients with respect to the target logit, and an activation extrapolation method, which uses an empirical distribution over the final hidden states to estimate the probability that $t$ will have the highest logit.

The authors show that importance sampling methods currently dominate activation extrapolation on all tasks, but provide strong arguments for why they might be relevant in more complex models, where finding any particular input that gives rise to a behavior is computationally intractable.

**Strengths:**

- The paper has a novel framing of low probability estimation as a concrete approach to understanding and improving worst-case model performance.  It is the first paper to explore extrapolation-based methods that don't require finding explicit examples of rare behaviors.


- The best performing method presented in the paper, MHIS, is motivated using existing work in automated redteaming. By adapting the Greedy Coordinate Gradient technique (previously used for finding jailbreaks) into a proposal distribution for Metropolis-Hastings sampling, the paper shows how adversarial attack methods can be leveraged for low-probability estimation.  This innovative connection between adversarial attacks and probability estimation shows how existing attack methods can be repurposed for studying the probability of rare behaviors on the normal input distribution.

**Weaknesses:**

- The models tested are small, and it's unclear whether these methods can be scaled up to actual models of interest.  However, the decision to use smaller models is justified, as it would be otherwise impossible to compute the true probability on all 2^32 input samples, which is necessary for measuring the performance of the methods.


- Given that adversarial training is stated as a part of the motivation for developing low probability estimation methods, there is a gap in the discussion around automated red teaming and adversarial training approaches for language models.  The related works section would benefit from an extended discussion on this area.  This context would help readers better understand how this work relates to and advances existing techniques for improving model robustness.

**Questions:**

- Do you have preliminary thoughts or analysis on how these methods might scale to larger language models?

---

> ### Author Response · Authors · 2024-11-20
> **Response to Reviewer DbYe**
>
> We are grateful to the reviewer for their generous review and comments.
>
> We agree that a significant limitation of this work is that the models we tested were all quite small. To answer the reviewer's question, we believe that some of our current methods -- in particular QLD and MHIS -- could work just as well on large models as they do for the 1- to 4-layer models. In particular, we have no reason to believe that the distributional assumption on the logit vector made by QLD is worse on larger models. Hopefully, future work can extend our methods to larger models with larger compute budgets. We are also excited about extending the methods to handle more realistic input distributions and more complex behaviors (beyond just a single-token output).
>
> Regarding the reviewer's comment on adversarial training, we have added some discussion in the related works section on Latent Adversarial Training (Casper et al., 2024; Sheshadri et al. 2024). Like activation extrapolation, LAT can work even when finding explicit inputs that cause the behavior is hard because it searches over activation space instead of input space.
>
> Stephen Casper, Lennart Schulze, Oam Patel, and Dylan Hadfield-Menell. Defending against unforeseen failure modes with latent adversarial training, 2024. URL https://arxiv.org/abs/2403.05030.
>
> Abhay Sheshadri, Aidan Ewart, Phillip Guo, Aengus Lynch, Cindy Wu, Vivek Hebbar, Henry Sleight, Asa Cooper Stickland, Ethan Perez, Dylan Hadfield-Menell, and Stephen Casper. Latent adversarial training improves robustness to persistent harmful behaviors in llms, 2024. URL https://arxiv.org/abs/2407.15549.

---

### Author Response · Authors · 2024-11-20
**Summary of changes**

We would like to thank all of the reviewers for their insightful comments. We have released a new version of the paper incorporating many suggestions. The main changes are:
- Added a discussion of Latent Adversarial Training (Casper et al., 2024; Sheshadri et al. 2024) in the Related Works section. Like activation extrapolation, LAT can work even when finding explicit inputs that cause the behavior is hard because it searches over activation space instead of input space.
- Added pseudocode to all of the algorithms in Appendix B.

Stephen Casper, Lennart Schulze, Oam Patel, and Dylan Hadfield-Menell. Defending against unforeseen failure modes with latent adversarial training, 2024. URL https://arxiv.org/abs/2403.05030.

Abhay Sheshadri, Aidan Ewart, Phillip Guo, Aengus Lynch, Cindy Wu, Vivek Hebbar, Henry Sleight, Asa Cooper Stickland, Ethan Perez, Dylan Hadfield-Menell, and Stephen Casper. Latent adversarial training improves robustness to persistent harmful behaviors in llms, 2024. URL https://arxiv.org/abs/2407.15549.

---

### Meta-Review · Area_Chair_Cy4r · 2024-12-23

**Metareview:**

All four reviewers recognize the importance and novelty of the paper’s core problem: estimating low-probability (rare) outputs of language models on a formally-specified input distribution. This problem is non-trivial and has significant implications for understanding worst-case model behaviors, improving adversarial training, and enhancing the robustness of language models.

Key Strengths:
	1.	Novel Problem Framing: The paper articulates the problem of low probability estimation (LPE) in language models, providing a clear motivation and framing it as a critical step towards mitigating rare but harmful outputs. This viewpoint offers a useful alternative perspective to standard adversarial training.
	2.	Methods and Empirical Results: The authors propose and evaluate multiple approaches—two based on importance sampling and two based on activation extrapolation. Reviewers appreciate the introduction of activation extrapolation as a novel class of methods. Although importance sampling methods currently outperform activation extrapolation, both classes of methods outperform naive random sampling, demonstrating the relevance of the proposed techniques.
	3.	Thorough Empirical Evaluation: The paper grounds its theoretical claims in empirical evaluations on small transformer models, where the exact ground-truth probability of certain outputs can be computed. While this is a simplified setting, the experiments are well-designed, and results show that even these first-step methods significantly improve upon naive baselines.
	4.	Clarity and Presentation Improvements: The authors have addressed reviewers’ requests by adding more discussion (e.g., on Latent Adversarial Training), incorporating pseudocode in the appendix, and clarifying the limitations and potential future directions.

**Additional Comments On Reviewer Discussion:**

Main Limitations and Responses:
	1.	Scalability and Complexity of Tasks: Reviewers noted that the proposed methods were tested only on small models and simplified settings (e.g., single-token outputs). The authors acknowledged these limitations, stating that their work should be viewed as a first step. They also point out that single-token proxies can be meaningful for certain tasks, and future research could extend these methods to larger models and more complex, multi-token behaviors.
	2.	Comparative Performance of Activation Extrapolation: Some reviewers highlighted that while activation extrapolation is intriguing, it currently underperforms importance sampling. The authors note that importance sampling can be seen as an instance of adversarial search and that future research might improve upon activation extrapolation or develop better LPE approaches to surpass these baselines.
	3.	Practical Applications and Black-Box Settings: There were concerns that these methods are mainly applicable in scenarios with white-box model access and may not directly translate to black-box settings. The authors clarified that the primary use-case is for model developers who have full access to their models (e.g., to improve safety before deployment). Still, the authors note that the conceptual framework can inspire techniques for more limited-access settings in the future.

Conclusion:

All reviewers improved their opinions after the author responses, with the majority expressing that the paper’s contributions, novelty, and well-grounded motivation justify its acceptance. While the paper’s methods are currently best suited as a starting point—especially for small models and single-token events—it is precisely such foundational work that paves the way for future innovation. The activation extrapolation approach, even if not currently outperforming importance sampling, adds conceptual breadth to the toolkit available for LPE and may inspire subsequent improvements.

---

### Decision · Program_Chairs · 2025-01-22

Accept (Spotlight)